# Entropy-Based Block Pruning for Efficient Large Language Models

**Liangwei Yang, Yuhui Xu** [*]**, Juntao Tan,**
**Doyen Sahoo**, **Silvio Savarese**, **Caiming Xiong**, **Huan Wang**, **Shelby Heinecke**
Salesforce AI Research

## Abstract

As large language models continue to scale, their growing computational and storage demands pose significant challenges for real-world deployment. In this work, we investigate redundancy within Transformer-based models and propose an entropy-based pruning strategy to enhance efficiency while maintaining performance. Empirical analysis reveals that the entropy of hidden representations decreases in the early blocks but progressively increases across most subsequent blocks. This trend suggests that entropy serves as a more effective measure of information richness within computation blocks. Unlike cosine similarity, which primarily captures geometric relationships, entropy directly quantifies uncertainty and information content, making it a more reliable criterion for pruning. Extensive experiments demonstrate that our entropy-based pruning approach surpasses cosine similarity-based methods in reducing model size while preserving accuracy, offering a promising direction for efficient model deployment.

## 1 Introduction

The emergence of large language models (LLMs) has reshaped current research landscape as well as empowering applications (Dubey et al., 2024; Team et al., 2024; Yang et al., 2025; Zhu et al., 2025). Scaling in size, they demonstrate remarkable performance across a wide range of domains/tasks such as chatbot (Achiam et al., 2023), code generation (Nijkamp et al., 2022), recommendation (Liang et al., 2025; Zhang et al., 2025), etc. Hidden behind these striking achievement, Transformer-based models (Waswani et al., 2017; Touvron et al., 2023; Jiang et al., 2023; Xue et al., 2024) scale their parameter size from millions to billions and research continues to explore even larger architectures (Liu et al., 2024) to further enhance their capabilities. However, the increasing scale in sizes result in substantial computational and storage costs, posing significant challenges for deployment.

Recent researches have detected the inherent redundancy of these pre-trained LLMs, especially on the layer level (Gromov et al., 2024; Men et al., 2024; Yang et al., 2024; Xu et al., 2022; Song et al., 2024; Chen et al., 2024; Kim et al., 2024). Models can maintain competitive performance even after a significant number of layers are removed, indicating that not all layers contribute equally. This observation has spurred extensive research on layer pruning techniques, which focus on eliminating redundant layers while retaining the model's core functionalities. LLMDrop (He et al., 2024) further discovered that the Attention block is more redundant than the MLP block, highlighting the need for a more fine-grained pruning approach to remove redundant components within each block rather than pruning entire layers. This redundancy provides new insights for optimizing model deployment, enabling more efficient acceleration strategies while maintaining performance.

For both layer and attention pruning, existing methods (Men et al., 2024; Yang et al., 2024; He et al., 2024; Mao et al., 2024) adhere to the practice of using cosine similarity to measure the redundancy between computation blocks. Redundant blocks with high similarity scores are identified and removed by comparing adjacent layers or selected layer pairs. However, cosine similarity primarily captures the geometric alignment of hidden representations, which does not necessarily reflect the actual information contribution of each layer. Consequently, relying solely on cosine similarity for pruning may lead to suboptimal decisions, potentially compromising model performance.

---

[*]Corresponding Author: yuhui.xu@salesforce.com

In this paper, we reconsider the use of cosine similarity as the criterion for pruning and propose **EntroDrop**, a novel approach that leverages entropy increase to assess the importance of computation blocks. Empirical analysis reveals that the entropy of hidden representations initially decreases in the early layers but progressively rises across subsequent layers. It suggests that entropy can serve as an effective indicator of information richness within each block. Unlike cosine similarity, which primarily captures geometric relationships, entropy directly quantifies the information content of a block's output, providing a more reliable basis for pruning decisions. Extensive experiments comparing entropy-based and cosine similarity-based pruning demonstrate that our entropy-driven approach more effectively preserves model accuracy while reducing computational costs. The code is open-sourced [1] to facilitate further research. Our key contributions are summarized as:

- We conduct an empirical analysis of entropy dynamics in hidden representations across LLM blocks during inference, offering new insights into information flow.
- We propose a novel entropy-based pruning strategy to effectively reduce model size and preserve performance.
- Extensive experiments demonstrate the superiority of EntroDrop over cosine similarity-based pruning methods.

## 2 PRELIMINARY

Transformer-based architectures consist of two primary computational blocks: the Attention and the MLP Block. They process hidden states and enrich them sequentially.

### 2.1 COMPUTATION BLOCKS

**Attention Block** enables each token in the input sequence to interact with others. Given an input $\mathbf{X}$, a Layer Normalization (LayerNorm) operation is applied before the self-attention computation $\mathbf{X}_{\text{norm}} = \text{LayerNorm}(\mathbf{X})$. Then, the attention mechanism computes as:

$$\mathbf{Y} = \text{Softmax}\left(\frac{\mathbf{Q}\mathbf{K}^T}{\sqrt{d_k}}\right)\mathbf{V},\tag{1}$$

where $\mathbf{Q} = \mathbf{X}_{\text{norm}}\mathbf{W}_Q$, $\mathbf{K} = \mathbf{X}_{\text{norm}}\mathbf{W}_K$, $\mathbf{V} = \mathbf{X}_{\text{norm}}\mathbf{W}_V$, and $\sqrt{d_k}$ is a scaling factor. The output $\mathbf{Y}$ represents the transformed hidden states.

**MLP Block** further transforms the output of Attention block. Assume the input for MLP block is also $\mathbf{X}$. It firstly applies layer normalization to stabilize the output as $\mathbf{X}_{\text{norm}} = \text{LayerNorm}(\mathbf{X})$. Then a two-layer feedforward network is calculated to process $\mathbf{X}_{\text{norm}}$ as:

$$\mathbf{Y} = \text{ReLU}(\mathbf{X}_{\text{norm}}\mathbf{W}_1 + \mathbf{b}_1)\mathbf{W}_2 + \mathbf{b}_2,\tag{2}$$

where $\mathbf{W}_1$, $\mathbf{W}_2$, $\mathbf{b}_1$ and $\mathbf{b}_2$ are learnable parameters. There are also other variants (Touvron et al., 2023) for this feedforward network. Together, the Attention Block and MLP Block form a complete **Transformer Block**, which can be stacked to build deep Transformer models. Each Transformer Block refines and enriches the hidden states, enabling hierarchical learning across multiple layers.

### 2.2 BLOCK-WISE PRUNING

Block-wise pruning aims to determine the importance of each computation block by analyzing the relationship between its input $\mathbf{X}$ and output $\mathbf{Y}$. The goal is to define an effective metric that identifies less informative blocks for removal while preserving essential model functionality. To quantify the importance of a block, an importance criterion is often calculated as:

$$I = g(\mathbf{X}, \mathbf{Y})\tag{3}$$

where $g(\cdot)$ is a function measuring the information contribution of the block and we prioritize the pruning on blocks with less importance score. No matter on which computation blocks, current methods (He et al., 2024; Men et al., 2024) judge the importance by cosine similarity and the importance criterion is calculated as $g(\mathbf{X}, \mathbf{Y}) = 1 - \frac{\mathbf{X} \cdot \mathbf{Y}}{|\mathbf{X}||\mathbf{Y}|}$. In this paper, we propose entropy increase, a

---

[1]https://github.com/SalesforceAIResearch/EntroDrop

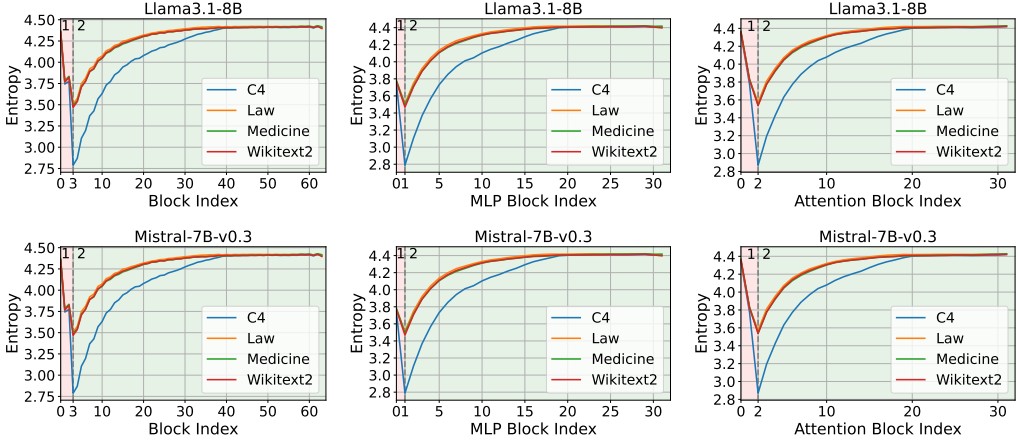

Figure 1: Entropy dynamics among layers during inference

new importance criterion based on empirical observations of entropy change across the layers. Entropy increase can better capture the information flow within the model, providing a more effective metric for identifying redundant blocks.

### 2.3 ENTROPY DEFINITION

For a discrete variable $\mathbf{Z}$ with probability mass function $p(z)$, its Shannon entropy is defined as:

$$H(\mathbf{Z}) = -\sum_z p(z) \log p(z),$$

which measures the average uncertainty or information content of $\mathbf{Z}$. Higher entropy indicates that $Z$ takes more uniformly distributed values (i.e., richer information), while lower entropy suggests more concentrated representations. In this work, we firstly discretize the hidden activations of large language models and compute their entropy layer by layer. This allows us to systematically observe how entropy evolves as the model depth increases, revealing a two-stage pattern of entropy decrease followed by entropy increase. This empirical finding serves as the foundation for our proposed entropy-based pruning strategy.

## 3 METHOD

### 3.1 OBSERVATIONS ON ENTROPY DYNAMICS

To investigate the entropy across different layers of Transformer models, we conduct experiments on Llama3.1-8B (Dubey et al., 2024) [2] and Mistral-7B-v0.3 (Jiang et al., 2023) [3]. We analyze the entropy trends during inference across Transformer Blocks, Attention Blocks, and MLP Blocks using four datasets: C4, Law, Medicine, and Wikitext2. We compute the entropy of hidden states at each block level and track its evolution across the network.

The experimental results, shown in Fig. 1, reveal a consistent two-stage behavior:

- **Stage 1: Entropy Decrease (Layers 1–3).** Early layers progressively reduce entropy, indicating strong information compression, noise filtering, and formation of compact representations.

- **Stage 2: Entropy Increase (Layers 3–32).** Subsequent layers gradually increase entropy, suggesting progressive contextual expansion and feature enrichment.

---

[2]https://huggingface.co/meta-llama/Llama-3.1-8B
[3]https://huggingface.co/mistralai/Mistral-7B-v0.3

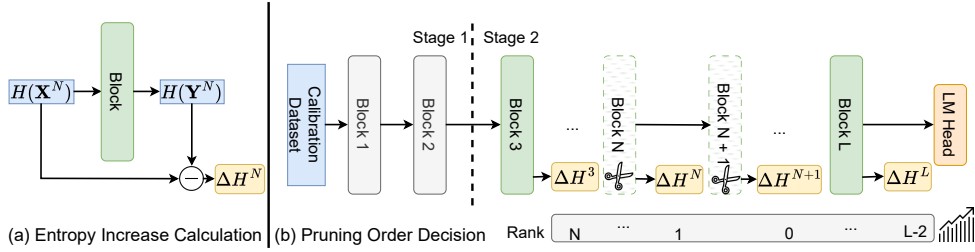

Figure 2: Overview of the EntroDrop framework. Stage 1 keeps intact, while Stage 2 exhibits increasing entropy. Blocks in Stage 2 are ranked based on their entropy increase, and those with the lower entropy increase are pruned earlier.

This pattern appears robust across all four datasets and aligns with prior findings (Yang et al., 2024; Men et al., 2024), which highlight the importance of early layers for information preservation. Our results further suggest that later layers contribute more uniformly to representation expansion, making them better candidates for pruning.

Entropy-lens Ali et al. (2025) studies entropy dynamics from prediction logits derived from hidden states, showing that different model families may exhibit distinct output-level entropy patterns. In contrast, we directly analyze the entropy of raw hidden representations, providing a more fine-grained view of information evolution inside the model. Across several mainstream LLMs—including Llama3.1-8B, Mistral-7B-v0.3, Qwen3-14B, and DeepSeek-V2-Lite-16B—we consistently observe an early-layer entropy decrease followed by a later-layer entropy increase (Appendix A.2). Although this two-stage trend is not universal across every architecture examined, it recurs across many widely used Transformer families, suggesting that such compression–expansion dynamics are a common emergent behavior in modern decoder-only LLMs.

## 3.2 ENTRODROP

Based on our empirical observations of entropy dynamics across Transformer models, we propose **EntroDrop**, a novel entropy-based pruning method that leverages entropy increase in later layers to identify and remove redundant computation blocks while preserving essential model performance. The framework is shown in Fig. 2.

We consider a pre-trained Transformer model consisting of $L$ computation blocks, each responsible for transforming hidden states as the input propagates through the network. Given a calibration dataset $\mathcal{D}$, we pass input samples through the model and collect the hidden states at each block:

$$\mathbf{Z}^l = f_l(\mathbf{Z}^{l-1}), \quad l = 1, 2, \ldots, L, \tag{4}$$

where $\mathbf{Z}^l$ represents the hidden state at the $l$-th block, and $f_l(\cdot)$ denotes the computation block function (e.g., Attention, MLP). Once the hidden states at all blocks are obtained, we estimate the entropy of each block and rank them according to entropy increases. The lowest $K$ blocks, which exhibit minimal entropy increase, are selected for pruning.

EntroDrop leverages a two-stage pruning strategy based on entropy observations. Stage 1 compresses the information, and no computation blocks are pruned in this stage. Stage 2 gradually increases the entropy, suggesting that these blocks perform similar hidden state enrichments. The transition point between the two stages, denoted as $S_{\text{start}}$, is determined using a calibration dataset.

To effectively estimate the importance of computation blocks, we define entropy increase as:

$$\Delta H^l = H(\mathbf{Z}^l) - H(\mathbf{Z}^{l-1}), \tag{5}$$

where $H(\cdot)$ represents the entropy estimation function. Blocks in Stage 2, indexed by $S_{\text{start}} \leq l \leq L$, are ranked in ascending order based on their entropy increase:

$$\text{Rank}(\Delta H^l) = \text{argsort}(\Delta H^l) \quad \text{for } l \geq S_{\text{start}}. \tag{6}$$

Finally, the $K$ blocks with the smallest entropy increase within Stage 2 are selected for pruning:

$$\mathcal{S}_{\text{prune}} = \{f_i \mid f_i \in \text{Rank}(\Delta H^l)_{S_{\text{start}}:L}[: K]\}, \tag{7}$$

| L | Method | Dataset | | | | | | | | | | | | | Average |
|---|---|---|---|---|---|---|---|---|---|---|---|---|---|---|---|
| | | PIQA | HellaSwag | WSC273 | CSQA | WinoGrande | ARC-E | ARC-C | OBQA | MMLU | CMMLU | RACE | XSum | GSM8k | |
| 0 | * | 0.803 | 0.6091 | 0.8864 | 0.5741 | 0.7388 | 0.7963 | 0.4898 | 0.3300 | 0.5908 | 0.3830 | 0.4086 | 0.0315 | 0.3715 | 0.5697 |
| 4 | LaCo | 0.5501 | 0.2975 | 0.6996 | 0.1196 | 0.6275 | 0.2908 | 0.2568 | 0.2400 | 0.1817 | 0.2070 | 0.2746 | 0.0284 | 0.0543 | 0.3216 |
| | ShortGPT | 0.7557 | 0.5458 | 0.8352 | 0.4808 | 0.7048 | 0.7104 | 0.4181 | 0.2620 | 0.4887 | 0.3598 | 0.3828 | 0.0291 | 0.0591 | 0.4662 |
| | Ours (Layer) | 0.7524 | 0.5467 | 0.8278 | 0.4865 | 0.7214 | 0.7079 | 0.4266 | 0.2700 | 0.4954 | 0.3458 | 0.3914 | 0.0445 | 0.0326 | 0.4730 |
| | LLMDrop | 0.8047 | 0.6051 | 0.8791 | 0.5717 | 0.7285 | 0.7971 | 0.4872 | 0.3380 | 0.5898 | 0.3828 | 0.3962 | 0.0238 | 0.3328 | 0.5484 |
| | Ours (Attn) | 0.8020 | 0.6062 | 0.8755 | 0.5725 | 0.7309 | 0.7984 | 0.4889 | 0.3380 | 0.5888 | 0.3819 | 0.4019 | 0.0371 | 0.3510 | **0.5568** |
| 8 | LaCo | 0.5952 | 0.3180 | 0.7033 | 0.2080 | 0.6377 | 0.3914 | 0.3029 | 0.1940 | 0.2802 | 0.2602 | 0.3062 | 0.0065 | 0.0078 | 0.3295 |
| | ShortGPT | 0.6627 | 0.3960 | 0.7143 | 0.5184 | 0.6598 | 0.5025 | 0.3294 | 0.2100 | 0.5086 | 0.3259 | 0.3167 | 0.0078 | 0.0099 | 0.3749 |
| | Ours (Layer) | 0.6627 | 0.3960 | 0.7143 | 0.5184 | 0.6598 | 0.5025 | 0.3294 | 0.2100 | 0.5086 | 0.3259 | 0.3167 | 0.0078 | 0.0099 | 0.3749 |
| | LLMDrop | 0.7998 | 0.5970 | 0.8718 | 0.5766 | 0.7364 | 0.7934 | 0.4753 | 0.3320 | 0.5917 | 0.3659 | 0.3952 | 0.0126 | 0.2813 | 0.5314 |
| | Ours (Attn) | 0.8003 | 0.5991 | 0.8681 | 0.5782 | 0.7332 | 0.7950 | 0.4765 | 0.3240 | 0.5902 | 0.3752 | 0.3952 | 0.0165 | 0.3495 | **0.5443** |
| 12 | LaCo | 0.5724 | 0.2937 | 0.6410 | 0.1630 | 0.5825 | 0.3013 | 0.2671 | 0.2020 | 0.2810 | 0.2214 | 0.2584 | 0.0032 | 0.0054 | 0.3110 |
| | ShortGPT | 0.5702 | 0.2795 | 0.6007 | 0.1974 | 0.5612 | 0.3367 | 0.2858 | 0.2080 | 0.2264 | 0.2426 | 0.2287 | 0.0083 | 0.0099 | 0.3014 |
| | Ours (Layer) | 0.6066 | 0.3415 | 0.6154 | 0.2424 | 0.5770 | 0.4146 | 0.2969 | 0.1820 | 0.3169 | 0.2595 | 0.3024 | 0.0062 | 0.0030 | 0.3339 |
| | LLMDrop | 0.7742 | 0.5614 | 0.8388 | 0.4054 | 0.7277 | 0.7483 | 0.4437 | 0.2820 | 0.5551 | 0.3143 | 0.3722 | 0.0139 | 0.0326 | 0.5071 |
| | Ours (Attn) | 0.7802 | 0.5749 | 0.8498 | 0.5446 | 0.7222 | 0.7546 | 0.4693 | 0.3080 | 0.5857 | 0.3636 | 0.3799 | 0.0147 | 0.0811 | **0.5255** |
| 16 | LaCo | 0.5577 | 0.2764 | 0.5165 | 0.2146 | 0.5367 | 0.3266 | 0.2509 | 0.1520 | 0.2637 | 0.2549 | 0.2641 | 0.0012 | 0.0023 | 0.2988 |
| | ShortGPT | 0.5403 | 0.2704 | 0.5421 | 0.1949 | 0.5501 | 0.3068 | 0.2619 | 0.1540 | 0.2367 | 0.2539 | 0.2488 | 0.0043 | 0.0059 | 0.2944 |
| | Ours (Layer) | 0.5272 | 0.2760 | 0.5275 | 0.1900 | 0.5067 | 0.2955 | 0.2491 | 0.1720 | 0.2473 | 0.2509 | 0.2411 | 0.0032 | 0.0020 | 0.2914 |
| | LLMDrop | 0.6926 | 0.4272 | 0.7875 | 0.2121 | 0.7017 | 0.5640 | 0.3328 | 0.2220 | 0.2735 | 0.2819 | 0.2938 | 0.0099 | 0.0126 | 0.4060 |
| | Ours (Attn) | 0.7514 | 0.4481 | 0.7656 | 0.3022 | 0.7048 | 0.6595 | 0.3925 | 0.2700 | 0.3586 | 0.2784 | 0.3282 | 0.0117 | 0.0511 | **0.4407** |

Table 1: Experiment Results on Mistral-7B-v0.3. $L$ indicates the number of pruned blocks.

where $\mathcal{S}_{\text{prune}}$ denotes the set of pruned blocks and $\Delta H^l$ represents the entropy increase of $l$ computation block. The bottom $k$ ranked layers are pruned to optimize efficiency. To estimate entropy efficiently, we explore multiple techniques:

- Bucket-based Estimation: Discretize activation values into bins and estimate based on frequency distribution.
- K-Nearest Neighbors (KNN): Computes entropy by estimating local density using KNN.
- Renyi Entropy: A generalization of Shannon entropy that provides a tunable parameter to control sensitivity to distribution variations.

Regardless of the estimation method used, entropy computation remains efficient. Our experimental results demonstrate that selecting an appropriate entropy estimation method is crucial for achieving optimal pruning performance. Among the approaches tested, Bucket-based estimation and KNN-based estimation were found to be particularly effective.

## 4 EXPERIMENTS

### 4.1 EXPERIMENTAL SETUP

**Models** We conduct experiments on two state-of-the-art decoder-only Transformer models: Llama3.1-8B and Mistral-7B-v0.3. To make a fair comparison, all experiments are finished on a single 40G A100 GPU device.

**Benchmarks** To evaluate the effectiveness of **EntroDrop**, we test on a diverse set of reasoning and comprehension benchmarks: Commonsense Reasoning: PIQA (Bisk et al., 2020), HellaSwag (Zellers et al., 2019), WSC273 (Sakaguchi et al., 2021), CSQA (Talmor et al., 2019), WinoGrande (Sakaguchi et al., 2021). Scientific and Knowledge-based QA: ARC-E (Clark et al., 2018), ARC-C (Clark et al., 2018), OBQA (Mihaylov et al., 2018). General and Subject-specific Knowledge: MMLU (Hendrycks et al., 2021b;a), CMMLU (Li et al., 2024), RACE (Lai et al., 2017). In addition, we include two generation-oriented benchmarks: XSum (Narayan et al., 2018) for abstractive summarization and GSM8K (Cobbe et al., 2021) for multi-step mathematical reasoning, allowing us to evaluate EntroDrop on both long-form and procedural generation tasks.

**Baselines** We compare **EntroDrop** against state-of-the-art pruning techniques in two categories: (1) Layer Pruning Methods that directly prune the whole transformer block: LaCo (Yang et al., 2024) and ShortGPT (Men et al., 2024). (2) Attention Pruning Method that only prunes the attention block: LLMDrop (He et al., 2024). These baselines allow us to assess how EntroDrop compares against existing pruning methods in terms of performance preservation under different pruning granularity.

| L | Method | Dataset | | | | | | | | | | | | | Average |
|---|--------|------|----------|--------|------|-----------|------|------|------|------|-------|------|------|-------|---------|
| | | PIQA | HellaSwag | WSC273 | CSQA | WinoGrande | ARC-E | ARC-C | OBQA | MMLU | CMMLU | RACE | XSum | GSM8k | |
| 0 | * | 0.7998 | 0.6003 | 0.8608 | 0.7166 | 0.7316 | 0.8148 | 0.5102 | 0.3340 | 0.6332 | 0.5090 | 0.3923 | 0.1302 | 0.5011 | **0.5872** |
| 4 | LaCo | 0.7628 | 0.5116 | 0.8059 | 0.6806 | 0.7103 | 0.7302 | 0.4462 | 0.2840 | 0.5949 | 0.4370 | 0.3761 | 0.1023 | 0.2857 | 0.5062 |
| | ShortGPT | 0.7557 | 0.5504 | 0.7949 | 0.6921 | 0.7017 | 0.7222 | 0.4420 | 0.3120 | 0.5802 | 0.4160 | 0.3818 | 0.1078 | 0.2942 | 0.5054 |
| | Ours (Layer) | 0.7573 | 0.5407 | 0.8242 | 0.7027 | 0.7088 | 0.7504 | 0.4275 | 0.2860 | 0.6212 | 0.4918 | 0.3818 | 0.1215 | 0.0834 | 0.5170 |
| | LLMDrop | 0.8025 | 0.5965 | 0.8352 | 0.7117 | 0.7498 | 0.8194 | 0.5188 | 0.3420 | 0.6312 | 0.5111 | 0.3933 | 0.1281 | 0.4754 | **0.5955** |
| | Ours (Attn) | 0.8003 | 0.6022 | 0.8498 | 0.7158 | 0.7364 | 0.8157 | 0.5179 | 0.3420 | 0.6238 | 0.5044 | 0.3895 | 0.1315 | 0.5057 | 0.5949 |
| 8 | LaCo | 0.6197 | 0.3098 | 0.6007 | 0.4005 | 0.6227 | 0.3952 | 0.2756 | 0.2360 | 0.4463 | 0.3405 | 0.2478 | 0.0280 | 0.0132 | 0.3093 |
| | ShortGPT | 0.6045 | 0.2825 | 0.5971 | 0.4046 | 0.5422 | 0.4289 | 0.2739 | 0.1820 | 0.3226 | 0.3153 | 0.2526 | 0.0310 | 0.0106 | 0.3015 |
| | Ours (Layer) | 0.6795 | 0.4384 | 0.7509 | 0.6216 | 0.6898 | 0.5644 | 0.3532 | 0.2120 | 0.5584 | 0.4408 | 0.3378 | 0.0837 | 0.0152 | 0.4583 |
| | LLMDrop | 0.7954 | 0.5877 | 0.8388 | 0.7174 | 0.7443 | 0.8119 | 0.5068 | 0.3560 | 0.6338 | 0.5073 | 0.4010 | 0.1268 | 0.4375 | **0.5932** |
| | Ours (Attn) | 0.7954 | 0.5921 | 0.8352 | 0.7183 | 0.7411 | 0.8186 | 0.5154 | 0.3540 | 0.6301 | 0.5041 | 0.3876 | 0.1263 | 0.4439 | 0.5909 |
| 12 | LaCo | 0.6202 | 0.3312 | 0.6337 | 0.1966 | 0.6219 | 0.4293 | 0.2705 | 0.1980 | 0.2428 | 0.2571 | 0.2813 | 0.0074 | 0.0014 | 0.3141 |
| | ShortGPT | 0.6007 | 0.3066 | 0.5861 | 0.5160 | 0.5501 | 0.4007 | 0.2765 | 0.1780 | 0.3605 | 0.3252 | 0.2660 | 0.0083 | 0.0015 | 0.3346 |
| | Ours (Layer) | 0.6007 | 0.3066 | 0.5861 | 0.5160 | 0.5501 | 0.4007 | 0.2765 | 0.1780 | 0.3605 | 0.3252 | 0.2660 | 0.0083 | 0.0015 | 0.3346 |
| | LLMDrop | 0.7867 | 0.5584 | 0.8608 | 0.6790 | 0.7253 | 0.7807 | 0.4753 | 0.3100 | 0.5992 | 0.4511 | 0.3799 | 0.1195 | 0.2002 | 0.5467 |
| | Ours (Attn) | 0.7867 | 0.5584 | 0.8608 | 0.6790 | 0.7253 | 0.7807 | 0.4753 | 0.3100 | 0.5992 | 0.4511 | 0.3799 | 0.1195 | 0.2002 | **0.5467** |
| 16 | LaCo | 0.5854 | 0.2904 | 0.6447 | 0.1957 | 0.5612 | 0.3443 | 0.2338 | 0.1600 | 0.2295 | 0.2527 | 0.2469 | 0.0024 | 0.0000 | 0.3071 |
| | ShortGPT | 0.5647 | 0.2754 | 0.5421 | 0.1949 | 0.5501 | 0.3194 | 0.2440 | 0.1540 | 0.2295 | 0.2529 | 0.2488 | 0.0023 | 0.0000 | 0.2956 |
| | Ours (Layer) | 0.5729 | 0.2705 | 0.5238 | 0.2113 | 0.5099 | 0.3165 | 0.2321 | 0.1380 | 0.2627 | 0.2538 | 0.2278 | 0.0145 | 0.0000 | 0.2980 |
| | LLMDrop | 0.6926 | 0.4272 | 0.7875 | 0.2121 | 0.7017 | 0.5640 | 0.3328 | 0.2220 | 0.2735 | 0.2819 | 0.2938 | 0.0595 | 0.1020 | 0.4207 |
| | Ours (Attn) | 0.7514 | 0.4481 | 0.7656 | 0.3022 | 0.7048 | 0.6595 | 0.3925 | 0.2700 | 0.3586 | 0.2784 | 0.3282 | 0.0936 | 0.1707 | **0.4603** |

Table 2: Experiment Results on Llama3.1-8B. $L$ indicates the number of pruned blocks. The best performance is marked in bold.

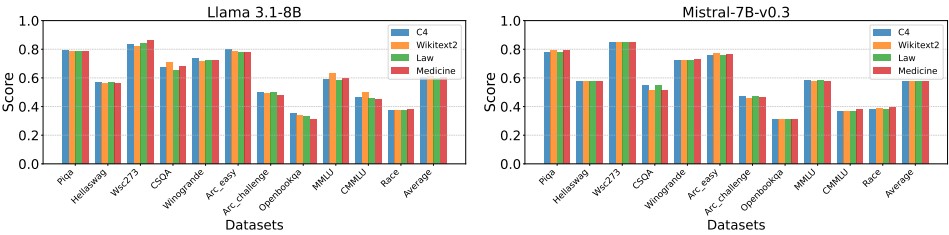

Figure 4: Impact of Calibration Datasets.

## 4.2 OVERALL PERFORMANCE

Our experimental results on Llama3.1-8B (Table 2) and Mistral-7B-v0.3 (Table 1) demonstrate the effectiveness of **EntroDrop**. We summarize the key findings as follows:

- **EntroDrop is effective across multiple models.** Our method consistently achieves the best performance across both Llama3.1-8B and Mistral-7B-v0.3. This demonstrates that EntroDrop is a generalizable pruning strategy applicable to different pre-trained LLMs.

- **EntroDrop outperforms both layer pruning and attention pruning baselines.** Compared to LaCo and ShortGPT (layer pruning) and LLMDrop (attention pruning), our method consistently achieves superior results. This suggests that our entropy-based metric effectively identifies and prunes redundant computation blocks at different granularities.

- **Pretrained Transformer models contain significant redundancy, especially in attention layers.** Our experiments show that removing up to 12 layers (37.5% of total attention layers) in Llama3.1-8B still retains over 95% of the model's original performance. This indicates that modern Transformers are often over-parameterized and that structured pruning can significantly improve efficiency without major performance degradation.

Overall, these findings confirm that entropy-based pruning is an effective and generalizable strategy for reducing redundant computation in large Transformer models. By leveraging entropy dynamics, **EntroDrop** enables efficient pruning while maintaining competitive performance.

## 4.3 IMPACT OF CALIBRATION DATASET

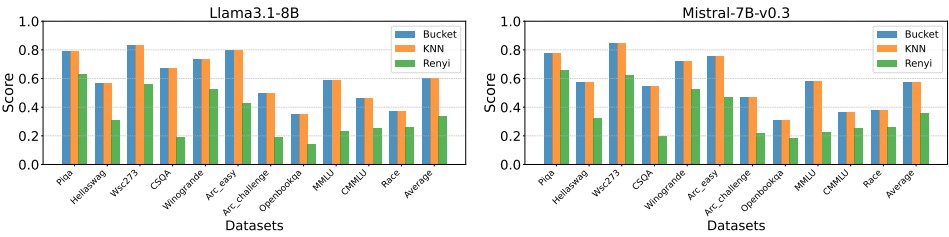

Figure 5: Impact of Entropy Estimate Methods.

Our pruning method relies on a **calibration dataset** to estimate entropy dynamics across Transformer layers. We investigate how different calibration datasets affect pruning results. Specifically, we evaluate two general-domain datasets (C4 (Raffel et al., 2020) and Wikitext (Merity et al., 2016)) and two specific-domain datasets (Medicine (Cheng et al., 2023) and Law (Cheng et al., 2023)).

Figure 3 presents the entropy increase heatmaps estimated using different calibration datasets on Llama3.1-

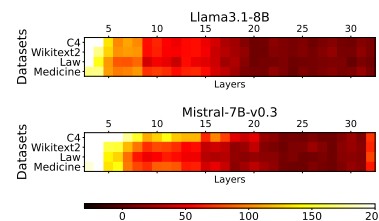

Figure 3: Calibration Datasets Heatmap

8B and Mistral-7B-v0.3. Across all models, entropy increase is smaller in deeper layers, indicating that these layers contribute less to new information processing and are more redundant. This suggests that deeper layers are natural candidates for pruning. Furthermore, despite differences in calibration datasets, the estimated entropy increase trends remain largely consistent. The relative importance of layers is preserved across general and domain-specific datasets, suggesting that our entropy-based pruning approach is robust to calibration dataset variations.

To further examine the impact of calibration datasets on model performance, Figure 4 presents the evaluation results of Llama3.1-8B and Mistral-7B-v0.3 after pruning 12 attention layers (37.5%). The results show that different calibration datasets lead to minimal differences in performance across all benchmark datasets, reinforcing the robustness of our entropy-based pruning strategy. Notably, even with domain-specific datasets (Medicine, Law), the average accuracy remains stable, indicating that the entropy estimation process generalizes well across different calibration datasets. These findings confirm that **EntroDrop** remains effective regardless of the calibration dataset, making it a flexible and generalizable pruning strategy.

## 4.4 ENTROPY ESTIMATION SENSITIVITY

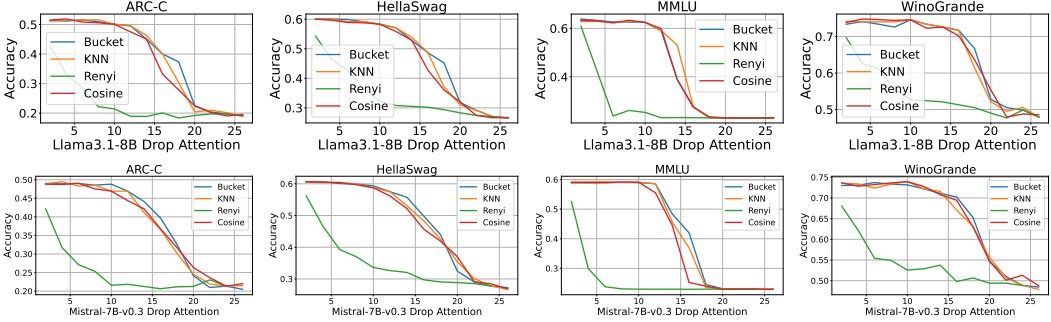

Figure 6: Attention Deletion Experiments

**Estimation Method**. Entropy estimation plays a crucial role in our pruning framework, as it directly influences the selection of redundant computation blocks. We evaluate three entropy estimation methods: Bucket, KNN and Renyi. To analyze the impact of different entropy estimation methods, we compare pruning results using these approaches on Llama3.1-8B and Mistral-7B-v0.3.

Figure 5 presents the evaluation results across multiple benchmark datasets when deleting 12 layers of attention blocks using our method. The results indicate that the choice of entropy estimation method significantly affects performance. Both Bucket-based and KNN-based estimation methods yield stable and high accuracy across all datasets, demonstrating their effectiveness in preserving essential model capabilities after pruning. In contrast, Renyi entropy estimation consistently underperforms, leading to noticeable accuracy degradation. This suggests that Renyi entropy may introduce excessive sensitivity to certain probability distributions, making it less suitable for pruning decisions of pre-trained Transformer blocks.

To investigate the redundancy in Transformer models, we analyze the impact of attention layer deletion across multiple datasets, including ARC-C, HellaSwag, MMLU, and WinoGrande. Figure 6 presents the performance degradation trend on MMLU as attention layers are progressively removed from Mistral-7B-v0.3. The results indicate that model performance remains stable until approximately 12 attention layers are removed, after which accuracy begins to degrade. This suggests that a significant portion of attention layers are redundant and can be pruned without substantial performance loss. Additionally, we compare different importance estimation methods for attention pruning. Both Bucket and KNN-based estimation methods consistently outperform Cosine Similarity, demonstrating their effectiveness in identifying unimportant attention layers. In contrast, Renyi entropy performs poorly, further confirming its limitations in guiding structured pruning.

**Estimation Hyper-parameter**. To further analyze the robustness of entropy estimation methods, we investigate the impact of different hyperparameter settings. We tune the following parameters: Bucket-based Estimation: Number of bins selected from $\{20, 40, 80, 160\}$. KNN-based Estimation: Number of nearest neighbors selected from $\{25, 50, 75, 100\}$. Figure 7 presents the estimated entropy values across Transformer layers using different hyperparameter settings. The results indicate that while different entropy estimation methods (Bucket vs. KNN) yield significantly different absolute entropy values, the relative importance ranking of layers remains largely unchanged within same estimation method. This suggests that the choice of hyperparameter (e.g., number of bins/neighbors) does not significantly impact the identification of redundant layers.

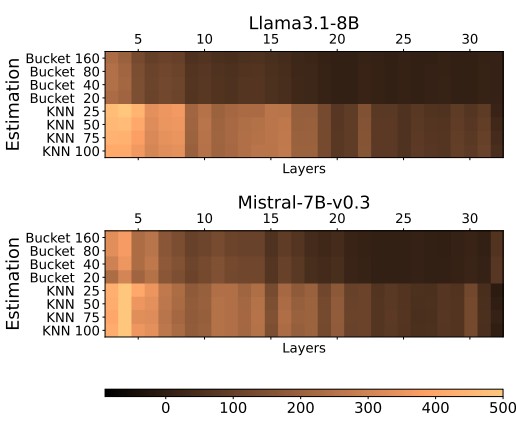

Figure 7: Hyperparameter Impact

The findings highlight the importance of selecting entropy estimation method while also reinforcing the stability of entropy-based pruning. Although different methods may compute varying absolute entropy values, the pruning decisions remain consistent. From our experiments, Bucket-based and KNN-based methods provide reliable performance, whereas using an inappropriate method like Renyi entropy could lead to suboptimal pruning outcomes.

## 4.5 SPEEDUP TEST

To evaluate the efficiency gains from pruning attention blocks, we conduct inference speed tests on Llama3.1-8B and Mistral-7B-v0.3. We prune attention layers progressively and measure both model performance and inference time. The speed test is performed by fixing the input sequence length to 1024 tokens and generating an output of 1024 tokens. Each experiment is repeated 10 times, and the average inference time is reported.

As shown in Fig. 8, inference time decreases nearly linearly with the number of pruned blocks, while accuracy remains stable for the first 12 pruned layers. Beyond this point, further pruning leads to noticeable degradation. Bucket- and KNN-based EntroDrop consistently outperform cosine-similarity pruning, offering better accuracy–speed trade-offs. Since EntroDrop removes entire attention blocks, the observed latency gains align closely with the theoretical FLOPs reduction (e.g., pruning 12 of 32 blocks leads to about 37.5% FLOPs reduction). These results demonstrate that EntroDrop yields substantial acceleration while preserving strong task performance.

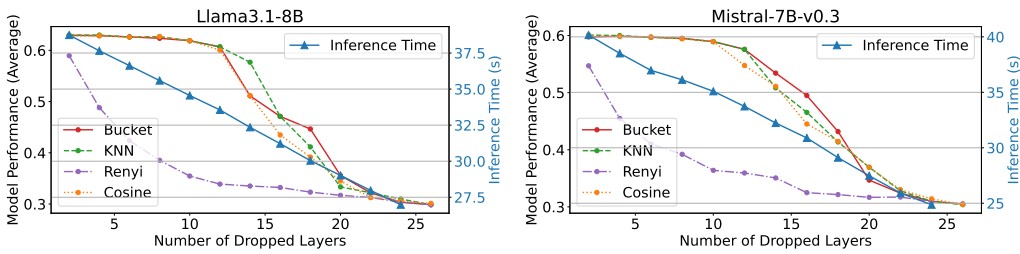

Figure 8: SpeedUp Experiments

## 5 RELATED WORK

**LLM Pruning.** In the era of large language models (LLMs), various methods have been proposed to reduce model size and accelerate inference (Frantar et al., 2022; Lin et al., 2024; Xiao et al., 2023; Shao et al., 2023; Zhu et al., 2023; Xu et al., 2023; Dettmers et al., 2023; Liu et al., 2023; Huang et al., 2025; Shen et al., 2025). Recent advances focus on post-training pruning techniques that eliminate redundant parameters or structures. (Gromov et al., 2024) empirically study the diminishing contribution of deeper layers in large Transformers and highlight the potential for layer pruning. While their work focuses on diagnosing redundancy, our method operationalizes this observation by quantifying information change through entropy and proposing a concrete, entropy-guided pruning strategy. SparseGPT (Frantar & Alistarh, 2023) leverages second-order information to identify unimportant parameters in LLMs. Wanda (Sun et al., 2023) introduces a pruning matrix that considers both weight magnitude and corresponding input activations. NEPENTHE (Liao et al., 2024) introduces a method that utilizes entropy to identify and remove low-entropy layers in deep neural networks, effectively reducing model depth while maintaining performance. E-Sparse (Li et al., 2023) introduces an entropy-based pruning method that enhances inference speed and reduces memory usage in large language models by leveraging information richness to guide N:M sparsity. SPP (Lu et al., 2024b) designs an efficient fine-tuning method to recover model performance post-pruning while maintaining sparsity. Beyond parameter pruning, structural pruning of LLMs has also gained popularity. LLM-Pruner (Ma et al., 2023) and ShearedLLaMA (Xia et al., 2023) remove unimportant structures such as layers and attention heads. Additionally, (Lu et al., 2024a) finds that certain experts in mixture-of-experts (MoE) LLMs can also be pruned. Among structural pruning methods, layer pruning is particularly relevant. Laco (Yang et al., 2024) reduces model depth by merging adjacent layers from the topmost layer downward. ShortGPT (Men et al., 2024) prunes unimportant layers based on a cosine similarity criterion. LLMDrop (He et al., 2024) finds that attention layers are more redundant than MLP layers but also relies on cosine similarity for pruning. Yang et al. (Yang et al.) also adopt an information-theoretic lens, using transfer entropy to measure how masking a block changes the model's output distribution. Their method evaluates output-level sensitivity via additional masked forward passes. In contrast, EntroDrop focuses on hidden-state entropy dynamics, requiring only a single unmasked forward pass and revealing a consistent compression–expansion pattern across layers. This representation-level perspective leads to a distinct and more efficient pruning criterion.

Different from prior approaches, we firstly analyze layer-wise entropy dynamics in LLMs and reveal a two-stage pattern of information compression followed by expansion. Based on this insight, we propose **EntroDrop**, which prunes blocks with the smallest entropy increase between layers.

## 6 CONCLUSION

In this paper, we present the first systematic study of layer-wise entropy dynamics in pretrained large language models. Our analysis reveals a two-stage pattern characterized by early-layer information compression followed by late-layer information enrichment. Building on this observation, we design **EntroDrop**, an entropy-based pruning framework that removes blocks contributing the least additional information. Extensive experiments show that EntroDrop achieves substantial parameter reduction and inference acceleration while maintaining high accuracy across multiple benchmarks.

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

# A APPENDIX

## A.1 THE USE OF LARGE LANGUAGE MODELS

We used large language models (LLMs) solely for grammar correction and language polishing of the manuscript. No part of the research ideation, experimental design, implementation, or analysis relied on LLMs. All methodological contributions and results were produced entirely by the authors.

## A.2 ENTROPY DYNAMICS OF DIFFERENT MODELS

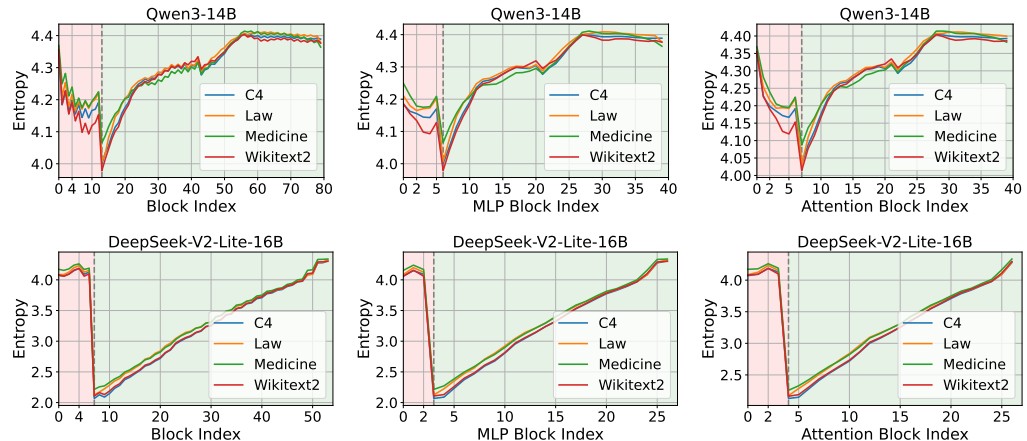

Figure 9: Entropy dynamics among layers during inference

To assess whether the observed entropy dynamics generalize beyond the two models reported in the main text, we further evaluate two widely used open-source LLMs: **Qwen3-14B** and **DeepSeek-V2-Lite-16B**. As shown in Fig. 9, both models exhibit a similar two-stage trend: an initial entropy *decrease* in the first few layers, followed by a gradual *increase* across the remaining depth. This pattern appears consistently across all four datasets. These results indicate that the compression–expansion transition is not specific to Llama- or Mistral-style architectures. While we also tested several additional models (not all shown), and the phenomenon is *not* universal, it is prominently present in many of the most commonly used open-source LLMs (Llama3, Mistral, Qwen, DeepSeek). This suggests that the entropy dynamics we report reflect a broadly shared behavior across mainstream Transformer models rather than an artifact of a single architecture.

Overall, the additional experiments reinforce our key conclusion: early layers tend to compress and stabilize representations, whereas later layers expand contextual information, making them more suitable for depth-based pruning.

## A.3 PRUNING EXPERIMENT RESULTS ON QWEN3-14B MODEL

As shown in Table 3, pruning Qwen3-14B under different budgets reveals clear differences among baseline methods. LaCo and ShortGPT degrade rapidly as $L$ increases, confirming that distance- and gradient-free heuristics are insufficient for identifying compressible layers in larger modern LLMs. LLMDrop (Attention) is consistently stronger, especially under higher pruning ratios. However, our method achieves the best performance across all pruning depths, with notable gains on challenging tasks such as WSC273, MMLU, GSM8k, and XSum. These results indicate that entropy-guided layer selection preserves essential reasoning and knowledge pathways more effectively than existing heuristics, leading to more robust model compression on Qwen-series architectures.

## A.4 POST-TRAINING EXPERIMENT

To further evaluate whether entropy-guided pruning preserves recoverable structure more effectively than cosine-similarity pruning, we conduct lightweight post-training experiments on the HellaSwag training split. For both Llama3.1-8B and Mistral-7B-v0.3, we prune the models with three budgets ($L = \{4, 8, 12\}$) and apply a short finetuning stage using only the HellaSwag training data. Since

| L | Method | PIQA | HellaSwag | WSC273 | CSQA | WinoGrande | ARC-E | ARC-C | OBQA | MMLU | CMMLU | RACE | XSum | GSM8k | Average |
|---|--------|------|-----------|--------|------|------------|-------|-------|------|------|-------|------|------|-------|---------|
| 0 | * | 0.8014 | 0.6090 | 0.8498 | 0.8018 | 0.7309 | 0.8422 | 0.5845 | 0.3520 | 0.7714 | 0.8019 | 0.4335 | 0.1196 | 0.8802 | 0.6599 |
| 4 | LaCo | 0.7220 | 0.4872 | 0.7875 | 0.6732 | 0.6985 | 0.7189 | 0.4573 | 0.2760 | 0.7564 | 0.7179 | 0.3770 | 0.0964 | 0.2183 | 0.5374 |
|  | ShortGPT | 0.7334 | 0.5309 | 0.7912 | 0.7830 | 0.6819 | 0.7319 | 0.4855 | 0.2740 | 0.7587 | 0.7364 | 0.4048 | 0.1143 | 0.3397 | 0.5666 |
|  | Ours (Layer) | 0.7704 | 0.5370 | 0.8168 | 0.7346 | 0.6559 | 0.7929 | 0.4684 | 0.3160 | 0.6299 | 0.6596 | 0.4077 | 0.1196 | 0.5648 | **0.5750** |
|  | LLMDrop (Attn) | 0.7949 | 0.6066 | 0.8315 | 0.7854 | 0.7182 | 0.8262 | 0.5759 | 0.3560 | 0.7644 | 0.7806 | 0.4316 | 0.1194 | 0.6027 | 0.6303 |
|  | Ours (Attn) | 0.7965 | 0.6062 | 0.8498 | 0.7879 | 0.7135 | 0.8333 | 0.5742 | 0.3400 | 0.7664 | 0.7884 | 0.4354 | 0.1171 | 0.8446 | **0.6503** |
| 8 | LaCo | 0.7367 | 0.4662 | 0.7070 | 0.4283 | 0.5983 | 0.7231 | 0.4053 | 0.2760 | 0.3583 | 0.3976 | 0.3550 | 0.1227 | 0.0235 | 0.4306 |
|  | ShortGPT | 0.6697 | 0.4188 | 0.7766 | 0.6626 | 0.6740 | 0.5219 | 0.3507 | 0.2300 | 0.7302 | 0.7217 | 0.3167 | 0.0308 | 0.0000 | 0.4695 |
|  | Ours (Layer) | 0.6872 | 0.4400 | 0.7179 | 0.6847 | 0.6306 | 0.6646 | 0.4019 | 0.2600 | 0.6977 | 0.7131 | 0.3943 | 0.1036 | 0.0341 | **0.4946** |
|  | LLMDrop (Attn) | 0.7775 | 0.5771 | 0.7985 | 0.7396 | 0.7245 | 0.8039 | 0.5538 | 0.3320 | 0.7598 | 0.6735 | 0.4067 | 0.1028 | 0.1054 | 0.5658 |
|  | Ours (Attn) | 0.7862 | 0.5717 | 0.8132 | 0.7486 | 0.7064 | 0.8308 | 0.5717 | 0.3480 | 0.7168 | 0.7080 | 0.4354 | 0.1161 | 0.6133 | **0.6128** |
| 12 | LaCo | 0.5604 | 0.2930 | 0.6374 | 0.2498 | 0.5841 | 0.3624 | 0.2790 | 0.1840 | 0.3126 | 0.2546 | 0.2555 | 0.0162 | 0.0000 | 0.3068 |
|  | ShortGPT | 0.6627 | 0.3866 | 0.6410 | 0.1957 | 0.5406 | 0.5972 | 0.2978 | 0.2200 | 0.2490 | 0.2646 | 0.2986 | 0.0898 | 0.0190 | 0.3433 |
|  | Ours (Layer) | 0.6464 | 0.3706 | 0.6007 | 0.4349 | 0.5635 | 0.5311 | 0.3106 | 0.2300 | 0.5755 | 0.5343 | 0.3167 | 0.0498 | 0.0008 | **0.3973** |
|  | LLMDrop (Attn) | 0.7476 | 0.5197 | 0.7582 | 0.2637 | 0.7048 | 0.7264 | 0.4795 | 0.2960 | 0.5862 | 0.2833 | 0.3579 | 0.0534 | 0.0106 | 0.4452 |
|  | Ours (Attn) | 0.7709 | 0.5090 | 0.7729 | 0.5807 | 0.6527 | 0.7567 | 0.4548 | 0.3260 | 0.5823 | 0.5875 | 0.3895 | 0.0909 | 0.0197 | **0.4995** |

Table 3: Experiment Results on Qwen3-14B. $L$ indicates the number of pruned blocks. The best performance (per block group) is marked in bold.

| Model | L | Base | Entropy (Ours) | | Recovery | Cosine | | Recovery |
|-------|---|------|--------|-----------|----------|--------|-----------|----------|
|  |  |  | Pruned | Finetuned | (%) | Pruned | Finetuned | (%) |
| Llama3.1-8B | 4 | 0.6003 | 0.5947 | 0.6002 | 98.8% | 0.5965 | 0.5953 | 30.0% |
|  | 8 | 0.6003 | 0.5921 | 0.5937 | 79.4% | 0.5877 | 0.5897 | 52.6% |
|  | 12 | 0.6003 | 0.5708 | 0.5836 | 42.5% | 0.5584 | 0.5594 | 4.6% |
| Mistral-7B-v0.3 | 4 | 0.6091 | 0.6062 | 0.6080 | 74.5% | 0.6051 | 0.6080 | 75.6% |
|  | 8 | 0.6091 | 0.5991 | 0.6045 | 46.3% | 0.5970 | 0.6036 | 62.4% |
|  | 12 | 0.6091 | 0.5749 | 0.5857 | 37.7% | 0.5614 | 0.5726 | 30.6% |

Table 4: Post-training recovery on HellaSwag for entropy-based pruning (ours) vs cosine similarity pruning. Base accuracy is shared across pruning methods.

all pruning methods begin from the same dense checkpoint, we report a shared baseline accuracy for each model and compare how much performance can be recovered after pruning. All results are evaluated on the HellaSwag test set, enabling a direct and controlled comparison between entropy-based and cosine-based criteria.

**Results.** Across both models, entropy-guided pruning shows markedly stronger recoverability than cosine similarity. For Llama3.1-8B, EntroDrop preserves most of the recoverable structure: pruning 4 layers leads to minimal degradation ($0.6003 \rightarrow 0.5947$), and post-training restores **98.8%** of the original accuracy, compared to cosine pruning which restores noticeably less. Even at larger budgets ($L = 8, 12$), EntroDrop-pruned models consistently regain a substantial portion of the lost accuracy (79.4% and 42.5% recovery), while cosine-pruned models exhibit weaker and more unstable recovery. Mistral-7B-v0.3 demonstrates the same pattern. Under EntroDrop, the model recovers **74.5%** of the gap at $L = 4$, and still restores 46.3% and 37.7% at $L = 8$ and $L = 12$, respectively—again exceeding the recovery obtained by cosine pruning. These results suggest that entropy reliably identifies layers whose removal can be compensated through small-scale post-training, indicating that entropy-based pruning preserves the model's intrinsic plasticity more effectively than cosine similarity, especially under deeper pruning budgets.

To evaluate whether post-training can recover the *general* capabilities of structurally pruned models, we design a controlled study that jointly varies the pruning criterion and the semantic relevance of post-training data. Experiments are conducted on two representative base models (Llama-3.1-8B and Mistral-7B), each pruned with a fixed 12-layer budget. To study capability recovery, we finetune each pruned model on **100k randomly sampled tuning examples** drawn from datasets with progressively increasing relevance to the evaluation tasks: (a) **Alpaca** (general instruction tuning, least related), (b) **Flan v2** (broad mixture of reasoning and knowledge tasks, moderately related),

Table 5: EntroDrop vs CosineDrop post-training results on 7 key benchmarks (Llama-3.1, 12-layer pruning).

| Task | Orig Base | Pruned EntroDrop | Cosine | Alpaca EntroDrop | Cosine | Flan v2 EntroDrop | Cosine | Mixed EntroDrop | Cosine |
|---|---|---|---|---|---|---|---|---|---|
| HellaSwag | 0.6003 | **0.5708** | 0.5584 | **0.5778** | 0.5643 | **0.5741** | 0.5599 | **0.5731** | 0.5588 |
| WinoGrande | 0.7316 | **0.7340** | 0.7253 | 0.7253 | **0.7261** | 0.7293 | **0.7324** | **0.7301** | 0.7245 |
| RACE | 0.3923 | 0.3703 | **0.3799** | **0.3837** | 0.3732 | **0.3780** | 0.3713 | 0.3770 | **0.3789** |
| CSQA | 0.7166 | 0.6740 | **0.6790** | 0.6388 | **0.6495** | **0.6978** | 0.6658 | **0.6978** | 0.6609 |
| ARC-E | 0.8148 | **0.7980** | 0.7807 | **0.7997** | 0.7824 | **0.7950** | 0.7811 | **0.7988** | 0.7824 |
| ARC-C | 0.5102 | **0.4957** | 0.4753 | **0.4940** | 0.4855 | **0.4889** | 0.4821 | **0.5000** | 0.4838 |
| OBQA | 0.3340 | **0.3540** | 0.3100 | **0.3460** | 0.3140 | **0.3500** | 0.3080 | **0.3540** | 0.3100 |
| Avg (7 tasks) | **0.5857** | **0.5710** | 0.5584 | **0.5665** | 0.5564 | **0.5733** | 0.5572 | **0.5758** | 0.5570 |

Table 6: EntroDrop vs Cosine pruned model post-training experiment across datasets (Mistral-7B, 12-layer pruning, 7-task subset).

| Task | Orig Base | Pruned EntroDrop | Cosine | Alpaca EntroDrop | Cosine | Flan v2 EntroDrop | Cosine | Mixed EntroDrop | Cosine |
|---|---|---|---|---|---|---|---|---|---|
| HellaSwag | 0.6091 | **0.5749** | 0.5614 | **0.5815** | 0.5686 | **0.5780** | 0.5645 | **0.5796** | 0.5678 |
| WinoGrande | 0.7388 | 0.7222 | **0.7277** | 0.7174 | **0.7253** | **0.7324** | 0.7309 | 0.7316 | **0.7411** |
| RACE | 0.4086 | **0.3799** | 0.3722 | **0.3856** | 0.3780 | 0.3799 | 0.3828 | 0.3732 | **0.3789** |
| CSQA | 0.5741 | **0.5446** | 0.4054 | **0.5127** | 0.3841 | **0.5152** | 0.3784 | **0.5651** | 0.4472 |
| ARC-E | 0.7963 | **0.7546** | 0.7483 | **0.7614** | 0.7593 | **0.7374** | 0.7260 | **0.7626** | 0.7500 |
| ARC-C | 0.4898 | **0.4693** | 0.4437 | **0.4556** | 0.4369 | **0.4556** | 0.4420 | **0.4599** | 0.4437 |
| OBQA | 0.3300 | **0.3080** | 0.2820 | **0.3060** | 0.2820 | **0.3240** | 0.2880 | **0.3020** | 0.2800 |
| Avg (7 tasks) | **0.5638** | **0.5502** | 0.5083 | **0.5315** | 0.5049 | **0.5310** | 0.5124 | **0.5380** | 0.5184 |

and (c) **Mixed** (most related), constructed by sampling from the training data of the evaluation benchmarks (HellaSwag, WinoGrande, RACE, CSQA, ARC, and OBQA). This yields a controlled semantic gradient (**Alpaca → Flan v2 → Mixed**) that allows us to study how recovery improves as post-training data becomes more aligned with the evaluation tasks. For each (criterion, dataset) configuration, we report per-task results, macro-average accuracy across all 11 tasks, and a direct comparison between EntropyDrop and CosineDrop under matched post-training conditions, using the Orig model as the common upper bound.

Across both Llama-3.1 (Table 5) and Mistral-7B (Table 6), the 7-task evaluation reveals a consistent pattern in post-pruning capability recovery. First, **EntropyDrop consistently outperforms cosine-based pruning** across all post-training settings, including the no-training case (PRUNED). This indicates that entropy-based saliency preserves functionally essential layers more effectively than representational similarity measures. Second, **the extent of recovery strongly depends on the semantic relevance of post-training data**. While Alpaca provides limited gains, Flan v2 yields moderate improvements, and the Mixed dataset—sampled from the training distributions of the evaluation tasks—produces the largest recovery. This alignment-driven trend holds for both pruning criteria and model architectures. Cosine-based pruning, however, exhibits a larger irrecoverable gap even under Mixed training. Third, **lightweight post-training (100k examples) provides substantial but alignment-dependent recovery**. When the finetuning data are semantically related to the evaluation tasks, performance improves markedly; however, unrelated datasets can introduce negative transfer and further degrade accuracy. Even under the best-aligned Mixed dataset, pruning-induced loss is only partially recoverable, indicating that post-training is highly effective but cannot fully compensate for information discarded by pruning.

We additionally evaluate post-training recovery under **8-layer Transformer pruning** on Llama-3.1-8B. Using the same setup as in the attention-pruning study, we finetune each pruned model with **100k examples** from three datasets with increasing semantic relevance (**Alpaca → Flan v2 → Mixed**). This allows us to examine how recovery depends on data alignment when entire Transformer layers are removed. Experiment results are shown in Table 7 Across all settings, **EntropyDrop again outperforms CosineDrop**—both immediately after pruning and after post-training—confirming the stability of entropy-guided layer selection. Recovery improves steadily with data relevance, but the pruning loss remains only partially recoverable, especially for cosine-based saliency.

Table 7: EntroDrop vs Cosine post-training results on 7 key benchmarks (Llama-3.1, 8-layer pruning).

| Task | Orig Base | Pruned (8L) EntroDrop | Cosine | Alpaca EntroDrop | Cosine | Flan v2 EntroDrop | Cosine | Mixed EntroDrop | Cosine |
|------|------|----------|--------|----------|--------|----------|--------|----------|--------|
| HellaSwag | 0.6003 | **0.4384** | 0.2825 | **0.4420** | 0.2903 | **0.4409** | 0.2866 | **0.4417** | 0.2922 |
| WinoGrande | 0.7316 | **0.6898** | 0.5422 | **0.6898** | 0.5414 | **0.7009** | 0.5430 | **0.6993** | 0.5446 |
| RACE | 0.3923 | **0.3378** | 0.2526 | **0.3445** | 0.2517 | **0.3349** | 0.2517 | **0.3426** | 0.2545 |
| CSQA | 0.7166 | **0.6216** | 0.4046 | **0.4963** | 0.4128 | **0.5872** | 0.4087 | **0.5512** | 0.4095 |
| ARC-E | 0.8148 | **0.5644** | 0.4289 | **0.5968** | 0.4411 | **0.5981** | 0.4297 | **0.5985** | 0.4335 |
| ARC-C | 0.5102 | **0.3532** | 0.2739 | **0.3592** | 0.2807 | **0.3575** | 0.2816 | **0.3490** | 0.2756 |
| OBQA | 0.3340 | **0.2120** | 0.1820 | **0.2640** | 0.1800 | **0.2640** | 0.1780 | **0.2600** | 0.1840 |
| **Avg (7 tasks)** | **0.5857** | **0.4596** | 0.3381 | **0.4561** | 0.3426 | **0.4691** | 0.3399 | **0.4632** | 0.3420 |

