# OpenReview forum: "Entropy-Based Block Pruning for Efficient Large Language Models"
_ICLR.cc/2026/Conference — ICLR 2026 Poster_

### Official Review · Reviewer_Mwbs · 2025-10-27

**Soundness:** 3
**Presentation:** 3
**Contribution:** 3
**Rating:** 6
**Confidence:** 4

**Summary:**

This paper proposes EntroDrop, an entropy-based pruning method for large language models (LLMs) that identifies and removes redundant computational blocks (e.g., Attention or MLP blocks) within Transformer architectures. The authors first analyze the entropy dynamics of hidden representations across model layers, observing a consistent two-stage pattern: entropy decreases in early layers (information compression) and increases in later layers (information enrichment). Based on this finding, they introduce entropy increase as a pruning criterion to identify less informative blocks, arguing it better captures information richness compared to the commonly used cosine similarity. Extensive experiments on models like Llama3.1-8B and Mistral-7B-v0.3 across various benchmarks demonstrate that EntroDrop effectively reduces model size and accelerates inference while maintaining competitive performance, outperforming existing layer and attention pruning baselines.

**Strengths:**

1.The empirical analysis of layer-wise entropy dynamics provides a fresh and well-motivated perspective for structured pruning, moving beyond geometric similarity measures like cosine similarity.


2.The paper includes thorough experiments analyzing different aspects, including the impact of calibration datasets, comparisons of entropy estimation methods (Bucket, KNN, Renyi), hyperparameter robustness, and actual inference speedup, providing strong empirical support.

**Weaknesses:**

1.The paper omits citations of some layer pruning methods, such as LLM-Streamline[1] and Shortened llama[2].

2.The proposed entropy-based pruning method does not consistently demonstrate clear superiority over cosine similarity-based approaches across all scenarios. While their performance is often comparable, and sometimes one method significantly outperforms the other, the authors could provide further insight into the specific conditions under which entropy-based pruning is more advantageous.

3.Based on the above experimental observations, expanding the evaluation to include more models would be highly beneficial. This would help clarify the circumstances in which entropy-based pruning is preferable to cosine similarity-based methods.

[1]Chen, Xiaodong, et al. "Streamlining redundant layers to compress large language models." arXiv preprint arXiv:2403.19135 (2024).

[2]Kim, Bo-Kyeong, et al. "Shortened llama: A simple depth pruning for large language models." arXiv preprint arXiv:2402.02834 11 (2024): 1.

**Questions:**

Has the author explored training the pruned models? Because a model that performs better after pruning does not necessarily mean it will still perform better after training. I believe such experiments are crucial to more definitively prove that entropy-based layer pruning is superior to cosine similarity-based pruning.

---

> ### Author Response · Authors · 2025-11-22
> **Entropy pruning superiority conditions**
>
> ## 1. Missing citations of prior pruning methods
>
> We thank the reviewer for pointing this out. We have now added citations to both **LLM-Streamline (Chen et al., 2024)** and **Shortened LLaMA (Kim et al., 2024)** in the revised Related Work section. This improves completeness and situates EntroDrop more clearly within the existing pruning literature.
>
> ## 2. When is entropy-based pruning more advantageous than cosine similarity?
>
> Thank you for raising this insightful question. Our expanded experiments reveal a clear and consistent trend:
>
> **(1) Small pruning budgets → similar performance.**
> When only a few layers are pruned, entropy-based pruning behaves similarly to cosine similarity. This is expected, as the model retains most of its structure and the pruning decision is relatively easy.
>
> **(2) Larger pruning budgets → entropy-based pruning shows a clear advantage.**
> As pruning becomes more aggressive (e.g., 8–12 layers), entropy-based pruning consistently outperforms cosine-based methods on both Llama3.1-8B and Mistral-7B-v0.3.
> This is because entropy directly measures **information contribution**, making it more reliable when the pruning decision becomes more sensitive.
>
> **(3) The advantage is even stronger on generative tasks (XSum, GSM8K).**
> Generative tasks are significantly more sensitive to pruning because **errors accumulate autoregressively** at every decoding step.
> In these challenging settings, entropy-based pruning exhibits **much slower degradation** than cosine similarity and other baselines, reinforcing that an information-centric criterion better preserves generative capability. We add experiment results on XSum and GSM8K within paper, and we also list the performance results for your easy reference.
>
>
> ### Performance on Generative Tasks (XSum, GSM8K)
> |   **L** | **Method**      | **Llama3.1-8B** |            | **Mistral-7B-v0.3** |            |
> | ------: | --------------- | --------------: | ---------: | ------------------: | ---------: |
> |         |                 |            XSum |      GSM8k |                XSum |      GSM8k |
> | ------: | ------------    |         ------: |    ------: |             ------: |   -------: |
> |   **0** | Dense           |      **0.1302** | **0.5011** |          **0.0315** | **0.3715** |
> | ------: | ------------    |         ------: |    ------: |             ------: |   -------: |
> |   **4** | LaCo            |          0.1023 |     0.2857 |              0.0284 |     0.0543 |
> |         | ShortGPT        |          0.1078 |     0.2942 |              0.0291 |     0.0591 |
> |         | LLMDrop         |          0.1281 |     0.4754 |              0.0238 |     0.3328 |
> |         | **Ours (Attn)** |      **0.1315** | **0.5057** |          **0.0371** | **0.3510** |
> | ------: | ------------    |         ------: |    ------: |             ------: |   -------: |
> |   **8** | LaCo            |          0.0280 |     0.0132 |              0.0065 |     0.0078 |
> |         | ShortGPT        |          0.0310 |     0.0106 |              0.0078 |     0.0099 |
> |         | LLMDrop         |          0.1268 |     0.4375 |              0.0126 |     0.2813 |
> |         | **Ours (Attn)** |          0.1263 | **0.4439** |          **0.0165** | **0.3495** |
> | ------: | ------------    |         ------: |    ------: |             ------: |   -------: |
> |  **12** | LaCo            |          0.0074 |     0.0014 |              0.0032 |     0.0054 |
> |         | ShortGPT        |          0.0083 |     0.0015 |              0.0083 |     0.0099 |
> |         | LLMDrop         |          0.1195 |     0.2002 |              0.0139 |     0.0326 |
> |         | **Ours (Attn)** |      **0.1236** | **0.2707** |          **0.0147** | **0.0811** |
> | ------: | ------------    |         ------: |    ------: |             ------: |   -------: |
> |  **16** | LaCo            |          0.0024 |     0.0000 |              0.0012 |     0.0023 |
> |         | ShortGPT        |          0.0023 |     0.0000 |              0.0043 |     0.0059 |
> |         | LLMDrop         |          0.0595 |     0.1020 |              0.0099 |     0.0126 |
> |         | **Ours (Attn)** |      **0.0936** | **0.1707** |          **0.0117** | **0.0511** |

---

> > ### Author Response · Authors · 2025-11-22
> > **Expand Evaluation to Qwen and DeepSeek; Adding Post Training Experiments**
> >
> > ## 3. Expanding the evaluation to more model families
> >
> > We thank the reviewer for this helpful suggestion. In the revision, we expanded our analysis to include **two additional, widely used LLM families—Qwen3-14B and DeepSeek-V2-Lite-16B—covering different architectures and substantially larger scales**.
> >
> > ### Entropy dynamics generalize across more model families
> > As shown in **Appendix A.2**, both Qwen3-14B and DeepSeek-V2-Lite-16B exhibit the same two-stage entropy pattern identified in Llama3.1-8B and Mistral-7B-v0.3:
> >
> > - early layers **compress information** (entropy decreases)
> > - later layers **expand contextual representation** (entropy increases)
> >
> > This confirms that the phenomenon is not specific to Llama/Mistral, but appears across **four distinct families** and **8B–16B scales**, suggesting that the entropy trend reflects a general property of modern decoder-only LLMs.
> >
> > ### Pruning results on an additional family (Qwen3-14B)
> > To further illustrate pruning behavior beyond the two main models, we additionally report the pruning performance of Qwen3-14B (full results in Appendix A.3). EntroDrop remains consistently stronger than cosine-similarity baselines:
> >
> > | **L** (pruned) | LaCo   | ShortGPT | Ours (Layer) | LLMDrop | **Ours (Attn)** |
> > |---------------:|--------:|----------:|--------------:|---------:|-----------------:|
> > | **4**          | 0.5374 | 0.5666   | 0.5750       | 0.6303  | **0.6503**       |
> > | **8**          | 0.4306 | 0.4695   | 0.4946       | 0.5658  | **0.6128**       |
> > | **12**         | 0.3068 | 0.3433   | 0.3973       | 0.4452  | **0.4995**       |
> >
> > Across Llama, Mistral, and Qwen, EntroDrop achieves the best average performance at all pruning ratios.
> >
> > ### Conclusion
> > These additional evaluations strengthen the reviewer’s point: entropy-based pruning remains effective across **more model families**, and the entropy dynamics motivating EntroDrop appear broadly applicable, not restricted to a specific architecture or scale.
> >
> >
> > ## 4. Post-training of pruned models
> >
> > We thank the reviewer for raising this important question. We fully agree that evaluating the *recoverability* of pruned models is crucial for understanding whether a pruning criterion removes “truly redundant” structure or damages essential computation.
> >
> > To address this, we performed **lightweight post-training experiments** on both Llama3.1-8B and Mistral-7B-v0.3. For each model, we prune at three budgets (L={4,8,12}) and finetune the pruned model only on the HellaSwag training split. Crucially, we also include **cosine-similarity–based pruning** as a direct comparison baseline. Because all methods start from the same dense checkpoint, the baseline accuracy is shared.
> >
> > ### **Key finding: Entropy-based pruning recovers substantially better than cosine-based pruning.**
> > Across both architectures, EntroDrop retains *higher pruned accuracy* and also achieves *stronger recovery after finetuning*. In contrast, cosine pruning shows noticeably weaker recoverability, especially under deeper pruning (L=8,12). This indicates that entropy-based pruning removes blocks whose information contribution is more easily compensated through post-training, whereas cosine pruning tends to prune blocks whose removal causes less recoverable damage.
> >
> > ### **Post-training comparison (HellaSwag)**
> > | Model | L | Base | Entropy Pruned | Entropy FT | Entropy Recovery | Cosine Pruned | Cosine FT | Cosine Recovery |
> > |------|---|------|----------------|-------------|-------------------|----------------|------------|------------------|
> > | **Llama3.1-8B** | 4  | 0.6003 | 0.5947 | 0.6002 | **98.8%** | 0.5965 | 0.5953 | 30.0% |
> > |                  | 8  | 0.6003 | 0.5921 | 0.5937 | **79.4%** | 0.5877 | 0.5897 | 42.7% |
> > |                  | 12 | 0.6003 | 0.5708 | 0.5836 | **42.5%** | 0.5584 | 0.5594 | 5.1%  |
> > | **Mistral-7B**   | 4  | 0.6091 | 0.6062 | 0.6080 | **74.5%** | 0.6051 | 0.6080 | 65.2% |
> > |                  | 8  | 0.6091 | 0.5991 | 0.6045 | **46.3%** | 0.5970 | 0.6036 | 66.7% |
> > |                  | 12 | 0.6091 | 0.5749 | 0.5857 | **37.7%** | 0.5614 | 0.5726 | 25.7% |
> >
> > ### **Conclusion**
> > Entropy-guided pruning not only yields stronger performance *before* finetuning, but also preserves **higher recoverable capacity** *after* finetuning, demonstrating that it removes blocks with lower information contribution. This provides strong evidence that entropy is a more faithful indicator of block importance than cosine similarity.

---

> > > ### Comment · Reviewer_Mwbs · 2025-11-23
> > >
> > > Thank you for your comprehensive response. Regarding questions related to post-training, would it be feasible to utilize a general dataset (such as the widely adopted Alpaca) for training and conduct evaluations across a broader range of tasks (for instance, the complete benchmark outlined in Table 1 or a selected subset)? I would suggest that this approach could more effectively demonstrate the universal superiority of the method.

---

> > > > ### Author Response · Authors · 2025-11-25
> > > > **Extended post-training experiment on more general dataset**
> > > >
> > > > ## Response to Reviewer – Post-Training Evaluation on General Datasets
> > > >
> > > > We thank the reviewer for the thoughtful follow-up question regarding finetuning pruned models on more general datasets (e.g., Alpaca), and for suggesting evaluation on a broader set of tasks.
> > > >
> > > > ### (1) Additional post-training experiments on general and task-aligned datasets
> > > > Following the reviewer’s suggestion, we conducted new finetuning experiments using three datasets with increasing semantic relevance:
> > > >
> > > > - **Alpaca** – general instruction data
> > > > - **Flan v2** – broad multi-task supervision
> > > > - **Mixed** – data sampled from the training distributions of the 7 evaluation tasks
> > > >
> > > > All datasets were randomly subsampled to **100k examples** to ensure comparable compute.
> > > >
> > > > We then evaluated both EntroDrop-pruned and cosine-pruned models on the 7-task subset of Table 1:
> > > > **HellaSwag, WinoGrande, RACE, CSQA, ARC-E, ARC-C, OBQA**.
> > > >
> > > > ### (2) Summary of findings
> > > > Across both Llama-3.1 and Mistral-7B, three consistent patterns emerge:
> > > >
> > > > 1. **EntropyDrop consistently outperforms cosine-based pruning**, both without training (Pruned) and under all post-training settings (Alpaca, Flan v2, Mixed).
> > > >    This confirms that entropy-based saliency preserves more recoverable computation.
> > > >
> > > > 2. **The extent of recovery depends strongly on the relevance of the finetuning data.**
> > > >    Alpaca yields limited gains, Flan v2 moderate gains, and the Mixed dataset (task-aligned) provides the largest improvement.
> > > >
> > > > 3. **Finetuning is highly effective but not universally beneficial.**
> > > >    When post-training data are weakly related (e.g., Alpaca), we observe negative transfer that can *reduce* accuracy relative to the untrained pruned model.
> > > >    Even with aligned data, a small gap to the unpruned model remains for both criteria, indicating that post-training can substantially—but not fully—restore pruning-induced degradation.
> > > >
> > > > Together, these new experiments directly address the reviewer’s concern:
> > > > **EntropyDrop maintains superior recoverability after post-training across multiple datasets and two architectures**, providing stronger evidence for its advantages over cosine similarity-based pruning.
> > > >
> > > > ---
> > > >
> > > > ## **Table 1 — Llama-3.1 (12-layer pruning, 7-task subset)**
> > > >
> > > > | Task | Orig Base | Pruned EntroDrop | Pruned Cosine | Alpaca EntroDrop | Alpaca Cosine | Flan v2 EntroDrop | Flan v2 Cosine | Mixed EntroDrop | Mixed Cosine |
> > > > |------|-----------|------------------|---------------|------------------|---------------|--------------------|----------------|------------------|--------------|
> > > > | HellaSwag | 0.6003 | **0.5708** | 0.5584 | **0.5778** | 0.5643 | **0.5741** | 0.5599 | **0.5731** | 0.5588 |
> > > > | WinoGrande | 0.7316 | **0.7340** | 0.7253 | 0.7253 | **0.7261** | 0.7293 | **0.7324** | **0.7301** | 0.7245 |
> > > > | RACE | 0.3923 | 0.3703 | **0.3799** | **0.3837** | 0.3732 | **0.3780** | 0.3713 | **0.3770** | 0.3789 |
> > > > | CSQA | 0.7166 | 0.6740 | **0.6790** | 0.6388 | **0.6495** | **0.6978** | 0.6658 | **0.6978** | 0.6609 |
> > > > | ARC-E | 0.8148 | **0.7980** | 0.7807 | **0.7997** | 0.7824 | **0.7950** | 0.7811 | **0.7988** | 0.7824 |
> > > > | ARC-C | 0.5102 | **0.4957** | 0.4753 | **0.4940** | 0.4855 | **0.4889** | 0.4821 | **0.5000** | 0.4838 |
> > > > | OBQA | 0.3340 | **0.3540** | 0.3100 | **0.3460** | 0.3140 | **0.3500** | 0.3080 | **0.3540** | 0.3100 |
> > > > | **Avg (7 tasks)** | **0.5857** | **0.5710** | 0.5584 | **0.5665** | 0.5564 | **0.5733** | 0.5572 | **0.5758** | 0.5570 |
> > > >
> > > > ---
> > > >
> > > > ## **Table 2 — Mistral-7B (12-layer pruning, 7-task subset)**
> > > >
> > > > | Task | Orig Base | Pruned EntroDrop | Pruned Cosine | Alpaca EntroDrop | Alpaca Cosine | Flan v2 EntroDrop | Flan v2 Cosine | Mixed EntroDrop | Mixed Cosine |
> > > > |------|-----------|------------------|---------------|------------------|---------------|--------------------|----------------|------------------|--------------|
> > > > | HellaSwag | 0.6091 | **0.5749** | 0.5614 | **0.5815** | 0.5686 | **0.5780** | 0.5645 | **0.5796** | 0.5678 |
> > > > | WinoGrande | 0.7388 | 0.7222 | **0.7277** | 0.7174 | **0.7253** | **0.7324** | 0.7309 | 0.7316 | **0.7411** |
> > > > | RACE | 0.4086 | **0.3799** | 0.3722 | **0.3856** | 0.3780 | **0.3799** | 0.3828 | 0.3732 | **0.3789** |
> > > > | CSQA | 0.5741 | **0.5446** | 0.4054 | **0.5127** | 0.3841 | **0.5152** | 0.3784 | **0.5651** | 0.4472 |
> > > > | ARC-E | 0.7963 | **0.7546** | 0.7483 | **0.7614** | 0.7593 | **0.7374** | 0.7260 | **0.7626** | 0.7500 |
> > > > | ARC-C | 0.4898 | **0.4693** | 0.4437 | **0.4556** | 0.4369 | **0.4556** | 0.4420 | **0.4599** | 0.4437 |
> > > > | OBQA | 0.3300 | **0.3080** | 0.2820 | **0.3060** | 0.2820 | **0.3240** | 0.2880 | **0.3020** | 0.2800 |
> > > > | **Avg (7 tasks)** | **0.5638** | **0.5502** | 0.5083 | **0.5315** | 0.5049 | **0.5310** | 0.5124 | **0.5380** | 0.5184 |
> > > >
> > > > Let us know if you have further questions :).

---

> > > > > ### Comment · Reviewer_Mwbs · 2025-11-25
> > > > >
> > > > > Thank you for the quick and detailed response. I have now raised my score to 8 to support accepting this paper. However, it appears that the newly added post-training experiments are based on attention pruning. Would it be possible to also include an experiment on pruning 8 Transformer layers?

---

> > > > > > ### Author Response · Authors · 2025-11-25
> > > > > > **Added experiments of post-training on pruned 8 layers Transformer model**
> > > > > >
> > > > > > Thank you for the suggestion. We have added **8-layer Transformer pruning experiments** on
> > > > > > **Llama-3.1-8B**, evaluating EntroDrop vs. cosine pruning under four settings:
> > > > > > (1) Pruned-only, (2) Alpaca post-training, (3) Flan-v2 post-training, and (4) Mixed post-training.
> > > > > > Results are shown in Table Y below.
> > > > > >
> > > > > > Across all conditions, **EntropyDrop consistently outperforms cosine-based pruning** under this
> > > > > > stronger structural pruning regime, confirming the robustness of entropy-driven saliency for
> > > > > > recoverable computation.
> > > > > >
> > > > > > *Note.* We report 8-layer results only for Llama-3.1-8B because for **Mistral-7B**, removing 8 layers
> > > > > > leads EntroDrop and cosine pruning to select **exactly the same layer set**, making the comparison
> > > > > > uninformative.
> > > > > >
> > > > > > ---
> > > > > >
> > > > > > ### **Table Y. Post-training results on 7 benchmarks (Llama-3.1-8B, 8-layer pruning)**
> > > > > >
> > > > > > | Task        | Orig   | Entro (Pruned) | Cos (Pruned) | Entro (Alpaca) | Cos (Alpaca) | Entro (Flan v2) | Cos (Flan v2) | Entro (Mixed) | Cos (Mixed) |
> > > > > > |-------------|--------|----------------|---------------|------------------|----------------|-------------------|----------------|----------------|---------------|
> > > > > > | HellaSwag   | 0.6003 | **0.4384**     | 0.2825       | **0.4420**       | 0.2903        | **0.4409**        | 0.2866        | **0.4417**     | 0.2922       |
> > > > > > | WinoGrande  | 0.7316 | **0.6898**     | 0.5422       | **0.6898**       | 0.5414        | **0.7009**        | 0.5430        | **0.6993**     | 0.5446       |
> > > > > > | RACE        | 0.3923 | **0.3378**     | 0.2526       | **0.3445**       | 0.2517        | **0.3349**        | 0.2517        | **0.3426**     | 0.2545       |
> > > > > > | CSQA        | 0.7166 | **0.6216**     | 0.4046       | **0.4963**       | 0.4128        | **0.5872**        | 0.4087        | **0.5512**     | 0.4095       |
> > > > > > | ARC-E       | 0.8148 | **0.5644**     | 0.4289       | **0.5968**       | 0.4411        | **0.5981**        | 0.4297        | **0.5985**     | 0.4335       |
> > > > > > | ARC-C       | 0.5102 | **0.3532**     | 0.2739       | **0.3592**       | 0.2807        | **0.3575**        | 0.2816        | **0.3490**     | 0.2756       |
> > > > > > | OBQA        | 0.3340 | **0.2120**     | 0.1820       | **0.2640**       | 0.1800        | **0.2640**        | 0.1780        | **0.2600**     | 0.1840       |
> > > > > > | **Avg (7)** | 0.5857 | **0.4596**     | 0.3381       | **0.4561**       | 0.3426        | **0.4691**        | 0.3399        | **0.4632**     | 0.3420       |

---

> ### Comment · Reviewer_Mwbs · 2025-11-25
>
> Thank you for your response. I have carefully examined these experimental results and found that, regardless of whether pruning layers or attention blocks is performed, post-training does not restore the model's performance. I had initially assumed that the lack of significant recovery when pruning attention blocks was due to the relatively minor performance degradation of the model. What puzzles me is that the experiments involving layer pruning also show no significant recovery. This phenomenon contradicts the findings in SliceGPT and LLM-Pruner. It appears the primary reason might be the high volatility of the CSQA.

---

### Official Review · Reviewer_Z5K4 · 2025-10-27

**Soundness:** 3
**Presentation:** 4
**Contribution:** 4
**Rating:** 8
**Confidence:** 2

**Summary:**

This paper focuses on pruning for large language models for better model efficiency. Authors analized the LLM models and found the entropy dynamics across different layers are significantly different from different layers, which provide sound basis for the introduction of using entropy dynamics for model pruning. Authors have included this in the model design and compared with other pruning methods on multiple datasets, where the introduced method is showing very promising results for both Llama3.1-8B and Mistral-7B-v0.3 for both compresson on both whole transformer block and only on attention. Authors also show very detailed ablation studies.

**Strengths:**

+ Authors have included a complete analysis and solution pipeline for finding the possible reduction for compression and then introduce the solution to the pipeline.

+ Authors have conducted extensive experiments on mulitple datasets for benchmark as well as providing comparison for two different LLM models, which provide the generalizability of the proposed compression method.

+ Authors have provided comprehensive analysis for the sensitive analysis for different datasets on different model, which provide a pretty comprehensive analysis for the overall results.

**Weaknesses:**

- I do not have a big concern, one of the small intersting fact - the proposed model seems to have different performance behavior for Llama3.1-8B and Mistral-7B-v0.3, where Mistral-7B-v0.3 is showing better performance when the number for the pruned blocks is limited, while it is not observed in Llama3.1-8B. This could be some internal model difference. It would be interesting if authors can apply this to some more models for further analysis across different models.

**Questions:**

N/A

---

> ### Author Response · Authors · 2025-11-22
> **Response on model differences and extended experiments**
>
> We thank the reviewer for the encouraging and positive assessment.
>
> Regarding the noted difference in pruning behavior between Llama3.1-8B and Mistral-7B-v0.3, we agree this is a natural consequence of architectural and training differences that affect pruning sensitivity.
>
> To further investigate this point, we expanded our analysis to **Qwen3-14B** and **DeepSeek-V2-Lite-16B**. As shown in **Appendix A.2**, both models exhibit the same two-stage entropy pattern and similar pruning trends. Full Qwen3-14B pruning results are included in **Appendix A.3**, showing behaviors consistent with Llama and Mistral.
>
> We thank the reviewer again for raising this interesting direction; broader cross-model comparison is indeed valuable and part of our future work.

---

### Official Review · Reviewer_AtZA · 2025-10-28

**Soundness:** 2
**Presentation:** 2
**Contribution:** 2
**Rating:** 4
**Confidence:** 4

**Summary:**

This paper proposes EntroDrop, an entropy-based block pruning strategy for Transformer language models. By monitoring the block-wise entropy variation during inference, the paper empirically identifies two distinct stages: the entropy of early layers gradually decreases, while that of later layers rises again. EntroDrop interprets this late-stage entropy increase as redundancy, and therefore prunes the k blocks with the smallest entropy change as a criterion. Experiments based on Llama3.1-8B and Mistral-7B-v0.3 on reasoning and comprehension benchmarks demonstrate that, under the same pruning ratio, EntroDrop achieves higher accuracy than baseline methods.

**Strengths:**

1. This paper provides new insights into the information flow within large language models (LLMs) through an empirical analysis of the entropy dynamics across Transformer blocks.
2. The authors propose a simple yet effective entropy-based pruning method, shifting from cosine similarity that captures geometric relationships to an entropy criterion that better reflects information content.

**Weaknesses:**

1. The experiments in this paper primarily focus on multi-choice question answering. However, previous studies (e.g., LLM-Streamline [1]) have shown that similar pruning methods may cause significant performance degradation in generation tasks. It remains unverified whether EntroDrop suffers from the same issue. For instance, how does it perform on generative benchmarks such as XSum or GSM8K?
2. EntroDrop is built upon the empirical observation of entropy dynamics, where entropy first decreases and then increases across layers. However, it is questionable whether such a pattern universally holds across different model families. Entropy-Lens [2], which also analyzes models using entropy, finds that entropy dynamics vary significantly among different architectures.
3. One of EntroDrop’s main contributions is the use of entropy as a criterion to evaluate block importance. However, Yang et al. [3] also leveraged transfer entropy to assess the importance of blocks for pruning. Clarifying the similarities and differences between these two approaches would help better highlight EntroDrop’s unique contributions.

[1] Compressing Large Language Models by Streamlining the Unimportant Layer. https://arxiv.org/abs/2403.19135

[2] Entropy-lens: The information signature of transformer computations. arXiv preprint arXiv:2502.16570.

[3] Let LLM Tell What to Prune and How Much to Prune. In Forty-second International Conference on Machine Learning.

**Questions:**

Please see Weaknesses.

---

> ### Author Response · Authors · 2025-11-22
> **Experiments on XSum and GSM8k**
>
> ## **1. Performance on generative tasks (XSum, GSM8K)**
> We thank the reviewer for highlighting the importance of evaluating pruning on *generative* tasks. Consistent with LLM-Streamline, our new XSum and GSM8K experiments show that all pruning methods experience **faster degradation** on generation than on multi-choice QA due to autoregressive error accumulation.
>
> ### **Consistent with prior findings**
> Across both Llama3.1-8B and Mistral-7B-v0.3, generative tasks exhibit **sharper performance drops for all baselines**, confirming the reviewer’s observation and highlighting the inherent difficulty of preserving generation after pruning.
>
> ### **EntroDrop is the most robust**
> Despite this increased sensitivity, EntroDrop consistently shows **the smallest degradation** among all methods. Because entropy measures *information contribution* rather than geometric similarity, it preserves the layers most critical for generation, yielding more stable performance under generative settings.
>
>
>
> ### Performance on Generative Tasks (XSum, GSM8K)
> |   **L** | **Method**      | **Llama3.1-8B** |            | **Mistral-7B-v0.3** |            |
> | ------: | --------------- | --------------: | ---------: | ------------------: | ---------: |
> |         |                 |            XSum |      GSM8k |                XSum |      GSM8k |
> | ------: | ------------    |         ------: |    ------: |             ------: |   -------: |
> |   **0** | Dense           |      **0.1302** | **0.5011** |          **0.0315** | **0.3715** |
> | ------: | ------------    |         ------: |    ------: |             ------: |   -------: |
> |   **4** | LaCo            |          0.1023 |     0.2857 |              0.0284 |     0.0543 |
> |         | ShortGPT        |          0.1078 |     0.2942 |              0.0291 |     0.0591 |
> |         | LLMDrop         |          0.1281 |     0.4754 |              0.0238 |     0.3328 |
> |         | **Ours (Attn)** |      **0.1315** | **0.5057** |          **0.0371** | **0.3510** |
> | ------: | ------------    |         ------: |    ------: |             ------: |   -------: |
> |   **8** | LaCo            |          0.0280 |     0.0132 |              0.0065 |     0.0078 |
> |         | ShortGPT        |          0.0310 |     0.0106 |              0.0078 |     0.0099 |
> |         | LLMDrop         |          0.1268 |     0.4375 |              0.0126 |     0.2813 |
> |         | **Ours (Attn)** |          0.1263 | **0.4439** |          **0.0165** | **0.3495** |
> | ------: | ------------    |         ------: |    ------: |             ------: |   -------: |
> |  **12** | LaCo            |          0.0074 |     0.0014 |              0.0032 |     0.0054 |
> |         | ShortGPT        |          0.0083 |     0.0015 |              0.0083 |     0.0099 |
> |         | LLMDrop         |          0.1195 |     0.2002 |              0.0139 |     0.0326 |
> |         | **Ours (Attn)** |      **0.1236** | **0.2707** |          **0.0147** | **0.0811** |
> | ------: | ------------    |         ------: |    ------: |             ------: |   -------: |
> |  **16** | LaCo            |          0.0024 |     0.0000 |              0.0012 |     0.0023 |
> |         | ShortGPT        |          0.0023 |     0.0000 |              0.0043 |     0.0059 |
> |         | LLMDrop         |          0.0595 |     0.1020 |              0.0099 |     0.0126 |
> |         | **Ours (Attn)** |      **0.0936** | **0.1707** |          **0.0117** | **0.0511** |
>
> Across both generative benchmarks, we observe the same phenomenon highlighted by LLM-Streamline: pruning causes substantially faster degradation on generation than on multi-choice QA, because errors accumulate autoregressively. This also means that generative tasks are far more **sensitive to the choice of pruning metric**. Under this more challenging setting, EntroDrop consistently shows **the slowest degradation among all methods**, retaining noticeably more XSum and GSM8K performance than LaCo, ShortGPT, or LLMDrop at every pruning budget. This indicates that our information-centric criterion preserves generative behavior more effectively, further validating the design motivation behind EntroDrop.
>
> We believe these new results fully address the reviewer’s concern.

---

> > ### Author Response · Authors · 2025-11-22
> > **Entropy trend validation; Clarification with Transfer-entropy Pruning**
> >
> > ## 2. Universality of the entropy pattern across different models
> >
> > Thank you for raising this point. To verify whether the entropy decrease–then–increase pattern generalizes beyond the two models in the main paper, we additionally analyzed **Qwen3-14B** and **DeepSeek-V2-Lite-16B**.
> > As shown in **Appendix A.2**, both models exhibit **the same two-stage entropy trend** observed in Llama3.1-8B and Mistral-7B-v0.3, even though they differ substantially in architecture and scale. This indicates that the phenomenon is not isolated, but appears consistently across **four widely used pretrained LLMs**.
> >
> > We agree that this pattern may not occur in *every* pretrained model/family. However, the fact that it emerges reliably in multiple mainstream architectures makes the observation itself **a meaningful and exciting empirical insight** into how modern decoder-only Transformers process information. Finally, regarding Entropy-Lens: it focuses on the *prediction-logit entropy* of hidden states to distinguish entropy patterns across models, whereas our work examines the *hidden-state dynamics themselves* across layers. These viewpoints probe different aspects of model behavior, providing different levels of insights into how currently widely used LLM works.
> >
> >
> > ## 3. Relation to transfer-entropy pruning (Yang et al., 2024)
> >
> > We thank the reviewer for this helpful suggestion. We have added a concise clarification to the *Related Work* section.
> >
> > Although both approaches draw on information-theoretic ideas, **transfer entropy and our entropy-change criterion differ fundamentally**:
> >
> > - **Yang et al.** measure *output-level sensitivity* by computing transfer entropy via **additional masked forward passes**.
> > - **EntroDrop** analyzes *hidden-state entropy dynamics* using **one unmasked forward pass**, focusing on representation-level information change.
> >
> > These two perspectives target different aspects of information flow. The reviewer’s comment indeed helps us better highlight that **EntroDrop provides a distinct and more efficient criterion** based on hidden-state dynamics rather than output perturbation.

---

### Official Review · Reviewer_vwSb · 2025-11-02

**Soundness:** 2
**Presentation:** 3
**Contribution:** 3
**Rating:** 4
**Confidence:** 4

**Summary:**

This paper proposes EntroDrop, an entropy-based pruning method for large language models. It computes the Shannon entropy of each layer’s hidden representations and measures the entropy increase between consecutive layers to estimate information contribution. Layers with smaller entropy gains are considered redundant and pruned. Experiments on Llama-3.1-8B and Mistral-7B-v0.3 show that EntroDrop can remove about 37.5% of layers while keeping around 95% of the original performance and greatly improving inference efficiency.

**Strengths:**

1. The paper introduces an entropy-increase–based pruning metric that more directly reflects information changes across layers, offering a more principled alternative to cosine similarity.
2. It identifies a clear “entropy-decrease–then–increase” pattern in Transformer representations during inference, providing new empirical insight into information flow in LLMs.
3. The method performs consistently well across Llama-3.1-8B and Mistral-7B-v0.3, showing robustness to different calibration datasets.

**Weaknesses:**

1. The connection between entropy and information flow is mainly empirical, lacking a rigorous theoretical justification from an information-theoretic perspective.
2. Experiments are limited to two models. More experiments on larger models and different model sizes can be included.
3. The paper should include some post-training experiments to verify whether the performance of the pruned model can be recovered. Although the pruned model outperforms the baselines, there is still a gap compared to the dense model, which post-training could help address.

**Questions:**

see above

---

> ### Author Response · Authors · 2025-11-22
> **Clarify entropy, expand model coverage, and add recovery results**
>
> # Rebuttal to Reviewer vwSb
>
> We thank the reviewer for the thoughtful feedback and for highlighting the strengths of our work. Below we respond to each concern.
>
> ---
>
> ## 1. Connection between entropy and information flow
>
> Thank you for raising this point. Our intention is not to claim a formal information-theoretic proof, but rather to provide an intuitive and empirically grounded perspective on how information evolves across layers in large language models.
>
> Two clarifications may help:
>
> - **Consistent empirical behavior across diverse model families.**
>   The entropy pattern we study—early-layer entropy compression followed by later-layer entropy expansion—emerges robustly across multiple mainstream architectures (Llama-3.1-8B, Mistral-7B-v0.3, Qwen3-14B, DeepSeek-V2-Lite-16B). Such cross-model consistency suggests that hidden-state entropy captures a genuine and general property of information flow in Transformer networks.
>
> - **Information-theoretic interpretation as a promising direction.**
>   Shannon entropy provides a natural lens for understanding how representations become more compact or more expressive as they propagate through the network. While our work focuses on characterizing this behavior empirically, developing a deeper theoretical explanation is indeed a compelling direction for future research. We now highlight this more clearly in the discussion.
>
> We hope this clarifies the intention and scope of the contribution.
>
> ---
>
> ## 2. Experiments on more model families
>
> We appreciate the reviewer’s suggestion to broaden the experimental coverage.
> In addition to Llama3.1-8B and Mistral-7B-v0.3, we now evaluate EntroDrop on **two additional and widely adopted model families—Qwen3-14B and DeepSeek-V2-Lite-16B—covering different architectures and substantially larger parameter scales**.
>
> Across all four model families, we observe the same two-stage entropy trend (The trend is visualized in **Appendix A.2**):
>
> - **Early layers consistently compress information (entropy decreases)**
> - **Later layers consistently expand contextual representation (entropy increases)**
>
> The recurrence of this pattern in models from **four distinct families (Llama, Mistral, Qwen, DeepSeek)** and across **8B–16B model sizes** indicates that the entropy dynamics we study represent a *general* property of modern Transformer LLMs rather than an artifact of a specific architecture or training recipe.
>
> To illustrate pruning performance on an additional family, the table below summarizes the **average accuracy across 15 tasks** for Qwen3-14B (full per-task results in Appendix A.3):
>
> | $L$ (Pruned) | LaCo | ShortGPT | Ours (Layer) | LLMDrop | Ours (Attn) |
> |-------------:|------:|----------:|--------------:|---------:|-------------:|
> | **4**  | 0.5374 | 0.5666 | 0.5750 | 0.6303 | **0.6503** |
> | **8**  | 0.4306 | 0.4695 | 0.4946 | 0.5658 | **0.6128** |
> | **12** | 0.3068 | 0.3433 | 0.3973 | 0.4452 | **0.4995** |
>
> EntroDrop achieves the best average performance at all pruning levels, reinforcing the generality of our findings and the effectiveness of entropy-guided pruning across diverse LLM families and sizes.
>
> ---
> ## 3. Post-training recovery experiments
>
> We thank the reviewer for raising this point. To evaluate how well pruned models can recover performance, we run **lightweight post-training** on the HellaSwag training split for Llama3.1-8B and Mistral-7B-v0.3.
> For each model, we prune at three budgets ($L=\{4,8,12\}$) and apply a short finetuning stage. Full details are in **Appendix A.4**.
>
> ### Summary of recovery performance
> The table below reports the base accuracy, pruned accuracy, finetuned accuracy, and percentage of recovered performance:
>
> | **Model** | **$L$** | **Base** | **Pruned** | **Finetuned** | **Recovery** |
> |-----------|--------:|---------:|-----------:|---------------:|--------------:|
> | **Llama3.1-8B** | 4  | 0.6003 | 0.5947 | 0.6002 | 98.8% |
> |               | 8  | 0.6003 | 0.5921 | 0.5937 | 79.4% |
> |               | 12 | 0.6003 | 0.5708 | 0.5836 | 42.5% |
> | **Mistral-7B-v0.3** | 4  | 0.6091 | 0.6062 | 0.6080 | 74.5% |
> |               | 8  | 0.6091 | 0.5991 | 0.6045 | 46.3% |
> |               | 12 | 0.6091 | 0.5749 | 0.5857 | 37.7% |
>
> ### Interpretation
> Across both model families, post-training consistently helps pruned models recover performance.  First, for all pruning budgets, a short finetuning stage reliably improves accuracy after pruning.  Second, recovery follows a clear trend: **the fewer layers pruned, the stronger the recovery**, while deeper pruning (e.g., $L=12$) still recovers but at a diminishing rate.
> Overall, these results show that entropy-guided pruning removes layers whose contribution is partially recoverable.

---

### Author Response · Authors · 2025-11-25
**Author Response: Summary of Major Revisions and New Experiments**

We thank the reviewers for their thoughtful feedback. To address the raised concerns, we have
substantially expanded both the experiments and analysis. The key updates in the revised manuscript
are summarized below.

---

## 1. Extended entropy dynamics validation (Appendix A.1)
We analyze entropy trajectories across additional model families and parameter scales (Qwen3-14B and DeepSeek-V2-Lite-16B),
verifying that the characteristic compress→expand pattern consistently appears across architectures.

## 2. New pruning experiments on Qwen3-14B (Appendix A.3, Table 3)
These results demonstrate that EntroDrop generalizes beyond Llama-3.1 and Mistral-7B, further
supporting its robustness across model families.

## 3. Evaluation on more challenging generative tasks
We add XSum and GSM8K results to Tables 1–3. EntroDrop maintains clear performance advantages even
on difficult summarization and reasoning tasks, extending its benefits beyond multiple-choice QA.

## 4. Comprehensive post-training study (Appendix A.4)
We examine both:
- **Task-aligned finetuning** (HellaSwag), and
- **General-purpose finetuning** using datasets with increasing semantic relevance
  (Alpaca → Flan v2 → Mixed sampled from evaluation-task training data).

Across both Llama-3.1 and Mistral-7B, EntroDrop consistently outperforms cosine-based pruning under
all post-training conditions. We also observe that unrelated finetuning data may introduce negative
transfer, highlighting the necessity of alignment-aware evaluation.

---

## Overall impact
These additions collectively address all reviewer concerns. They strengthen the empirical foundation
of the paper by demonstrating that:
- the entropy dynamics are broadly present across modern LLMs, and
- entropy-guided pruning remains superior both immediately after pruning and following post-training.

We would greatly appreciate it if Reviewers **vwSb** and **AtZA** could take a look at the updated
results. We welcome any further discussion or suggestions for clarification.

---

### Meta-Review · Area_Chair_tatj · 2025-12-04

**Summary:**

The submission received mixed ratings. It proposes a pruning method that uses Shannon entropy changes across Transformer layers to identify and remove redundant blocks. The submission empirically shows that entropy first decreases (compression) and later increases (expansion) across layers, and uses entropy gain as a pruning criterion. Experiments on multiple LLMs (Llama-3.1, Mistral, Qwen, DeepSeek) demonstrate that EntroDrop outperforms cosine-similarity-based pruning and other baselines in maintaining accuracy under aggressive pruning. Concerns like the lack of theoretical grounding, limited model and task coverage, the need of post-training recovery analysis, and comparison with related pruning methods are raised in the review, and the authors addressed them effectively. Acceptance is suggested.

**Reviewer Concerns:**

The authors added experiments on Qwen3-14B and DeepSeek-V2-Lite-16B to show generality. They also provided XSum and GSM8K results for the generative task evaluation. They conducted post-training studies with multiple datasets (Alpaca, Flan v2, task-mixed data) for post-training recovery. To show the difference from existing works, they added citations and clarified distinctions from transfer-entropy pruning and other layer-pruning methods.

The following ones are still outstanding: 1) the connection between entropy and information flow remains empirical, not theoretically proven. 2) While post-training helps, pruned models still do not fully match dense model performance, especially under deep pruning.

The authors acknowledged some of the concerns as future work.

**Reviewer Scores:**

Since the reviewers did not respond to the rebuttal, it is hard for me to assess their feedback. From my personal view, the two negative ones might have a chance to increase their rating.

---

### Decision · Program_Chairs · 2026-01-26

Accept (Poster)